# Towards a Unified Framework of Clustering-based Anomaly Detection

## Abstract

Unsupervised Anomaly Detection (UAD) plays a crucial role in identifying abnormal patterns within data without labeled examples, holding significant practical implications across various domains. Although the individual contributions of representation learning and clustering to anomaly detection are well-established, their interdependencies remain under-explored due to the absence of a unified theoretical framework. Consequently, their collective potential to enhance anomaly detection performance remains largely untapped. To bridge this gap, in this paper, we propose a novel probabilistic mixture model for anomaly detection to establish a theoretical connection among representation learning, clustering, and anomaly detection. By maximizing a novel anomaly-aware data likelihood, representation learning and clustering can effectively reduce the adverse impact of anomalous data and collaboratively benefit anomaly detection. Meanwhile, a theoretically substantiated anomaly score is naturally derived from this framework. Lastly, drawing inspiration from gravitational analysis in physics, we have devised an improved anomaly score that more effectively harnesses the combined power of representation learning and clustering. Extensive experiments, involving 17 baseline methods across 30 diverse datasets, validate the effectiveness and generalization capability of the proposed method, surpassing state-of-the-art methods.

## 1 Introduction

Unsupervised Anomaly Detection (UAD) refers to the task dedicated to identifying abnormal patterns or instances within data in the absence of labeled examples [8]. It has long received extensive attention in the past decades for its wide-ranging applications in numerous practical scenarios, including financial auditing [3], healthcare monitoring [44] and e-commerce sector [23]. Due to the lack of explicit label guidance, the key to UAD is to uncover the dominant patterns that widely exist in the dataset so that samples do not conform to these patterns can be recognized as anomalies. To achieve this, early works [7] have heavily relied on powerful unsupervised *representation learning* methods to extract the normal patterns from high-dimensional and complex data such as images, text, and graphs. More recent works [45, 2] have utilized *clustering*, a widely observed natural pattern in real-world data, to provide critical global information for anomaly detection and achieved tremendous success.

While the individual contributions of representation learning and clustering to anomaly detection are well-established, their interrelationships remain largely unexplored. Intuitively, *discriminative representation learning* can leverage accurate clustering results to differentiate samples from distinct clusters in the embedding space (i.e., ①). Similarly, it can utilize accurate anomaly detection to avoid preserving abnormal patterns (i.e., ②). For *accurate clustering*, it can gain advantages from representation learning by operating in the discriminative embedding space (i.e., ③). Meanwhile, it

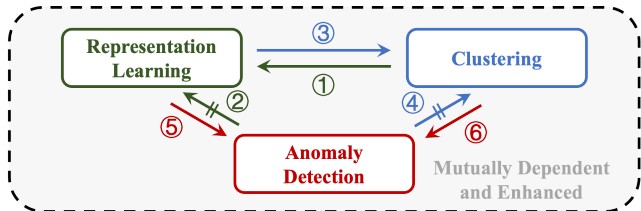

Figure 1: Interdependent relationships among representation learning, clustering, and anomaly detection.

can potentially benefit from accurate anomaly detection by excluding anomalies when formulating clusters (i.e., ④). *Anomaly detection* can greatly benefit from both discriminative representation learning and accurate clustering (i.e., ⑤ & ⑥). However, these benefits hinge on the successful identification of anomalies and the reduction of their detrimental impact on the aforementioned tasks. As depicted in Figure 1, the integration of these three elements exhibits a significant reciprocal nature. In summary, representation learning, clustering, and anomaly detection are interdependent and intricately intertwined. Therefore, it is crucial for anomaly detection to *fully leverage and mutually enhance the relationships among these three components*.

Despite the intuitive significance of the interactions among representation learning, clustering, and anomaly detection, existing methods have only made limited attempts to exploit them and fall short of expectations. On one hand, some methods [58] have acknowledged the interplay among these three factors, but their focus remains primarily on the interactions between two factors at a time, making only targeted improvements. For instance, some strategies include explicitly removing outlier samples during the clustering process [9] or designing robust representation learning methods [10] to mitigate the influence of anomalies. On the other hand, recent methods [45] have begun to explore the simultaneous optimization of these three factors within a single framework. However, these attempts are still in the stage of merely superimposing the objectives of the three factors without a unified theoretical framework. This lack of a guiding framework prevents the adequate modeling of the interdependencies among these factors, thereby limiting their collective contribution to a unified anomaly detection objective. Consequently, we aim to address the following question: *Is it possible to employ a unified theoretical framework to jointly model these three interdependent objectives, thereby leveraging their respective strengths to enhance anomaly detection?*

In this paper, we try to answer this question and propose a novel model named UniCAD for anomaly detection. The proposed UniCAD integrates representation learning, clustering, and anomaly detection into a unified framework, achieved through the theoretical guidance of maximizing the anomaly-aware data likelihood. Specifically, we explicitly model the relationships between samples and multiple clusters in the representation space using the probabilistic mixture models for the likelihood estimation. Moreover, we creatively introduce a learnable indicator function into the objective of maximum likelihood to explicitly attenuate the influence of anomalies on representation learning and clustering. Under this framework, we can theoretically derive an anomaly score that indicates the abnormality of samples, rather than heuristically designing it based on clustering results as existing works do. Furthermore, building upon this theoretically supported anomaly score and inspired by the theory of universal gravitation, we propose a more comprehensive anomaly metric that considers the complex relationships between samples and multiple clusters. This allows us to better utilize the learned representations and clustering results from this framework for anomaly detection.

To sum up, we underline our contributions as follows:

• We propose a unified theoretical framework to jointly optimize representation learning, clustering, and anomaly detection, allowing their mutual enhancement and aid in anomaly detection.

• Based on the proposed framework, we derive a theoretically grounded anomaly score and further introduce a more comprehensive score with the vector summation, which fully releases the power of the framework for effective anomaly detection.

• Extensive experiments have been conducted on 30 datasets to validate the superior unsupervised anomaly detection performance of our approach, which surpassed the state-of-the-art through comparative evaluations with 17 baseline methods.

## 2   Related Work

Typical unsupervised anomaly detection (UAD) methods calculate a continuous score for each sample to measure its anomaly degree. Various UAD methods have been proposed based on different assumptions, making them suitable for detecting various types of anomaly patterns, including subspace-based models [24], statistical models [16], linear models [49, 32], density-based models [6, 38], ensemble-based models [39, 29], probability-based models [40, 58, 28, 27], neural network-based models [42, 51], and cluster-based models [18, 9]. Considering the field of anomaly detection has progressed by integrating clustering information to enhance detection accuracy [26, 56], we primarily focus on and analyze anomaly patterns related to clustering, incorporating a global clustering perspective to assess the degree of anomaly. Notable methods in this context include CBLOF [18], which evaluates anomalies based on the size of the nearest cluster and the distance to the nearest large cluster. Similarly, DCFOD [45] introduces innovation by applying the self-training architecture of the deep clustering [50] to outlier detection. Meanwhile, DAGMM [58] combines deep autoencoders with Gaussian mixture models, utilizing sample energy as a metric to quantify the anomaly degree. In contrast, our approach introduces a unified theoretical framework that integrates representation learning, clustering, and anomaly detection, overcoming the limitations of heuristic designs and the overlooked anomaly influence in existing methods.

## 3   Methodology

In this section, we first define the problem we studied and the notations used in this paper. Then we elaborate on the proposed method UniCAD. More specifically, we first introduce a novel learning objective that optimizes representation learning, clustering, and anomaly detection within a unified theoretical framework by maximizing the data likelihood. A novel anomaly score with theoretical support is also naturally derived from this framework. Then, inspired by the concept of universal gravitation, we further propose an enhanced anomaly scoring approach that leverages the intricate relationship between samples and clustering to detect anomalies effectively. Finally, we present an efficient iterative optimization strategy to optimize this model and provide a complexity analysis for the proposed model.

**Definition 1** (Unsupervised Anomaly Detection). *Given a dataset* $\mathbf{X} \in \mathbb{R}^{N \times D}$ *comprising* $N$ *instances with* $D$*-dimensional features, unsupervised anomaly detection aims to learn an anomaly score* $o_i$ *for each instance* $\mathbf{x}_i$ *in an unsupervised manner so that the abnormal ones have higher scores than the normal ones.*

### 3.1   Maximizing Anomaly-aware Likelihood

Previous research has demonstrated the importance of discriminative representation and accurate clustering in anomaly detection [45]. However, the presence of anomalous samples can significantly disrupt the effectiveness of both representation learning and clustering [12]. While some existing studies have attempted to integrate these three separate learning objectives, the lack of a unified theoretical framework has hindered their mutual enhancement, leading to suboptimal results.

To tackle this issue, in this paper, we propose a unified and coherent approach that considers representation learning, clustering, and anomaly detection by maximizing the likelihood of the observed data. Specifically, we denote the parameters of representation learning as $\Theta$, the clustering parameter as $\Phi$, and the dynamic indicator function for anomaly detection as $\delta(\cdot)$. These parameters are optimized simultaneously by maximizing the likelihood of the observed data $\mathbf{X}$:

$$\max \log p(\mathbf{X}|\Theta, \Phi) = \max \sum_{i=1}^{N} \delta(\mathbf{x}_i) \log p(\mathbf{x}_i|\Theta, \Phi) = \max \sum_{i=1}^{N} \delta(\mathbf{x}_i) \log \sum_{k=1}^{K} p(\mathbf{x}_i, c_i = k|\Theta, \Phi),$$
(1)

where $c_i$ represents the latent cluster variable associated with $\mathbf{x}_i$, and $c_i = k$ denotes the probabilistic event that $\mathbf{x}_i$ belongs to the $k$-th cluster. The $\delta(\mathbf{x}_i)$ is an indicator function that determines whether a sample $\mathbf{x}_i$ is an anomaly of value 0 or a normal sample of value 1.

### 3.1.1  Joint Representation Learning and Clustering with $p(\mathbf{x}_i|\Theta, \Phi)$

Based on the aforementioned advantages of MMs, we estimate the likelihood $p(\mathbf{x}_i|\Theta, \Phi)$ with mixture models defined as:

$$p(\mathbf{x}_i|\Theta, \Phi) = \sum_{k=1}^{K} p(\mathbf{x}_i, c_i = k|\Theta, \Phi) = \sum_{k=1}^{K} p(c_i = k) \cdot p(\mathbf{x}_i|c_i = k, \Theta, \boldsymbol{\mu}_k, \Sigma_k)$$
$$= \sum_{k=1}^{K} \omega_k \cdot p(\mathbf{x}_i|c_i = k, \Theta, \boldsymbol{\mu}_k, \Sigma_k), \tag{2}$$

where $\Phi = \{\omega_k, \boldsymbol{\mu}_k, \Sigma_k\}$. The mixture model is parameterized by the prototypes $\boldsymbol{\mu}_k$, covariance matrices $\Sigma_k$, and mixture weights $\omega_k$ from all clusters. $\sum_{k=1}^{K} \omega_k = 1$, and $k = 1, 2, \cdots, K$.

In practice, the samples are usually attributed to high-dimensional features and it is challenging to detect anomalies from the raw feature space [41]. Therefore, modern anomaly detection methods [42, 58] often map raw data samples $\mathbf{X} = \{\mathbf{x}_i\} \in \mathbb{R}^{N \times D}$ into a low-dimensional representation space $\mathbf{Z} = \{\mathbf{z}_i\} \in \mathbb{R}^{N \times d}$ with a representation learning function $\mathbf{z}_i = f_\Theta(\mathbf{x}_i)$ and detect anomalies within this latent representation space.

Following this widely adopted practice, we model the distribution of samples in the latent representation space with a multivariate Student's-$t$ distribution giving its cluster $c_i = k$. The Student's-$t$ distribution is robust against outliers due to its heavy tails. Bayesian robustness theory leverages such distributions to dismiss outlier data, favoring reliable sources, making the Student's-$t$ process preferable over Gaussian processes for data with atypical information [1]. Thus the probability distribution of generating $\mathbf{x}_i$ with latent representation $\mathbf{z}_i$ given its cluster $c_i = k$ can be expressed as:

$$p(\mathbf{x}_i|c_i = k, \Theta, \boldsymbol{\mu}_k, \Sigma_k) = \frac{\Gamma(\frac{\nu+1}{2})|\Sigma_k|^{-1/2}}{\Gamma(\frac{\nu}{2})\sqrt{\nu\pi}} \left(1 + \frac{1}{\nu} D_M(\mathbf{z}_i, \boldsymbol{\mu}_k)^2\right)^{-\frac{\nu+1}{2}}, \tag{3}$$

where $\mathbf{z}_i = f_\Theta(\mathbf{x}_i)$ denotes the representation obtained from the data mapped through the neural network parameterized by $\Theta$. $\Gamma$ denotes the gamma function while $\nu$ is the degree of freedom. $\Sigma_k$ is the scale parameter. $D_M(\mathbf{z}_i, \boldsymbol{\mu}_k) = \sqrt{(\mathbf{z}_i - \boldsymbol{\mu}_k)^T \Sigma_k^{-1} (\mathbf{z}_i - \boldsymbol{\mu}_k)}$ represents the Mahalanobis distance [33]. In the unsupervised setting, as cross-validating $\nu$ on a validation set or learning it is unnecessary, $\nu$ is set as 1 for all experiments [50, 48]. The overall marginal likelihood of the observed data $\mathbf{x}_i$ can be simplified as:

$$p(\mathbf{x}_i|\Theta, \Phi) = \sum_{k=1}^{K} \omega_k \cdot \frac{\pi^{-1} \cdot |\Sigma_k|^{-1/2}}{1 + D_M(\mathbf{z}_i, \boldsymbol{\mu}_k)^2}. \tag{4}$$

### 3.1.2  Anomaly Indicator $\delta(\mathbf{x}_i)$ and Score $o_i$

As we have discussed, the indicator function $\delta(\mathbf{x}_i)$ not only benefits both representation and clustering but also directly serves as the output of anomaly detection. Ideally, with the percentage of outliers denoted as $l$, an optimal solution for $\delta(\mathbf{x}_i)$ that maximizes the objective function $J(\Theta, \Phi)$ entails setting all $\delta(\mathbf{x}_i) = 0$ for $\mathbf{x}_i$ among the $l$ percent of outliers with lowest generation possibility $p(\mathbf{x}_i|\Theta, \Phi)$, and otherwise $\delta(\mathbf{x}_i) = 1$ is set for the remaining normal samples. Therefore, the indicator function is determined as:

$$\delta(\mathbf{x}_i) = \begin{cases} 0, & \text{if } p(\mathbf{x}_i|\Theta, \Phi) \text{ is among the } l \text{ lowest,} \\ 1, & \text{otherwise.} \end{cases} \tag{5}$$

As this method involves sorting the samples based on the generation probability as being anomalous, the values of $p(\mathbf{x}_i|\Theta, \Phi)$ can serve as a form of anomaly score, a classic approach within the mixture model framework [40, 58]. This suggests that the likelihood of a sample being anomalous is inversely related to its generative probability since a lower generative probability indicates a higher chance of the sample being an outlier. Thus the anomaly score of sample $\mathbf{x}_i$ can be defined as:

$$o_i = \frac{1}{p(\mathbf{x}_i|\Theta, \Phi)} = \frac{1}{\sum_{k=1}^{K} \omega_k \cdot \frac{\pi^{-1} \cdot |\Sigma_k|^{-1/2}}{1 + D_M(\mathbf{z}_i, \boldsymbol{\mu}_k)^2}}. \tag{6}$$

## 3.2 Gravity-inspired Anomaly Scoring

In practical applications, it is proved that anomaly scores derived from generation probabilities often yield suboptimal performance [17]. This observation prompts a reconsideration of *how to fully leverage the complex relationships among samples or even across multiple clusters for anomaly detection*. In this section, we first provide a brief introduction to the concept of Newton's Law of Universal Gravitation [35] and then demonstrate how the anomaly score is intriguingly similar to this cross-field principle. Finally, we discuss the advantages of introducing the vector sum operation into the anomaly score inspired by the analogy.

### 3.2.1 Analog Anomaly Scoring and Force Analysis

To begin with, Newton's Law of Universal Gravitation [35] stands as a fundamental framework for describing the interactions among entities in the physical world. According to this law, every object in the universe experiences an attractive force from another object. In classical mechanics, force analysis involves calculating the vector sum of all forces acting on an object, known as the **resultant force**, which is crucial in determining an object's acceleration or change in motion:

$$\vec{\mathbf{F}}_{i,\text{total}} = \sum_{k=1}^{K} \vec{\mathbf{F}}_{ik}, \quad \text{with } \vec{\mathbf{F}}_{ik} = \frac{G \cdot m_i m_k}{r_{ik}^2} \cdot \vec{\mathbf{r}}_{ik}, \tag{7}$$

where $\vec{\mathbf{F}}_{ik}$ represents the $k$-th force acting on the object $i$. This force is proportional to the product of their masses, ($m_i$ and $m_k$), and inversely proportional to the square of the distance $r_{ik}$ between them. $G$ represents the gravitational constant, and $\vec{\mathbf{r}}_{ij}$ is the unit direction vector.

Similarly, if denoting: $\widetilde{\mathbf{F}}_{ik} = p(\mathbf{x}_i, c_i = k|\Theta, \Phi) = \omega_k \cdot \frac{\pi^{-1} \cdot |\Sigma_k|^{-1/2}}{1 + D_M(\mathbf{z}_i, \boldsymbol{\mu}_k)^2}$, the score of Equation (6) bears analogies to the summation of the magnitudes of forces as:

$$o_i = \frac{1}{\sum_{k=1}^{K} \widetilde{\mathbf{F}}_{ik}}, \quad \text{with } \widetilde{\mathbf{F}}_{ik} = \frac{\widetilde{G} \cdot \widetilde{m}_i \widetilde{m}_k}{\widetilde{r}_{ik}^2}, \tag{8}$$

where $\widetilde{G} = \pi^{-1}$, $\widetilde{m}_k = \omega_k |\Sigma_k|^{-1/2}$, $\widetilde{m}_i = 1$, and $\widetilde{r}_{ik} = \sqrt{1 + D_M(\mathbf{z}_i, \boldsymbol{\mu}_k)^2}$. Here, $\widetilde{r}_{ik}$ is taken as the measure of distance within the representation space, modified slightly by an additional term for smoothness. The constant $\widetilde{G}$ serves a role akin to the gravitational constant in this analogy, whereas $\widetilde{m}_k$ resembles the concept of mass for the cluster. The notation $\widetilde{m}_i$ suggests a standardization where the mass of each data point is considered uniform and not differentiated.

### 3.2.2 Anomaly Scoring with Vector Sum

Comparing Equation (7) with Equation (8), what still differs is that, unlike a simple sum of the scalar value, the resultant force $\vec{\mathbf{F}}_{i,\text{total}}$ employs the vector sum and incorporates both the magnitude and direction $\widehat{\mathbf{r}}_{ik}$ of each force. This distinction is crucial because forces in different directions can neutralize each other with a large angle between them or enhance each other's effects with a small angle. Inspired by this difference, we consider modeling the relationship between samples and clusters as a vector, and aggregating them through vector summation. The vector-formed anomaly score $o_i^V$ is defined as:

$$o_i^V = \frac{1}{\| \sum_{k=1}^{K} \widetilde{\mathbf{F}}_{ik} \cdot \vec{\mathbf{r}}_{ik} \|}, \tag{9}$$

where $\vec{\mathbf{r}}_{ik}$ represents the unit direction vector in the representation space from the sample $\mathbf{z}_i$ to the cluster prototype $\boldsymbol{\mu}_k$, and $\| \cdot \|$ represents the $L_2$ norm.

### 3.3 Iterative Optimization

Given the challenge posed by the interdependence of the parameters of the network $\Theta$ and those of the mixture model $\{\omega_k, \boldsymbol{\mu}_k, \Sigma_k\}$ in joint optimization, we propose an iterative optimization procedure. The pseudocode for training the model is presented in Algorithm 1 in the appendix.

#### 3.3.1 Update $\Phi$

To update the parameters of the mixture model $\Phi = \{\omega_k, \boldsymbol{\mu}_k, \Sigma_k\}$, we use the Expectation-Maximization (EM) algorithm to maximize equation (1) [36]. The detailed derivation is included in Appendix B.

**E-step.** During the E-step of iteration $(t+1)$, our goal is to compute the posterior probabilities of each data point belonging to the $k$-th cluster within the mixture model. Given the observed sample $\mathbf{x}_i$ and the current estimates of the parameters $\Theta^{(t)}$ and $\Phi^{(t)}$, the expected value of the likelihood function of latent variable $c_k$, or the posterior possibilities, can be expressed as:

$$\boldsymbol{\tau}_{ik}^{(t+1)} = p(c_i = k | \mathbf{x}_i, \Theta, \Phi^{(t)}) = \frac{p(\mathbf{x}_i, c_i = k | \Theta, \Phi^{(t)})}{\sum_{j=1}^{K} p(\mathbf{x}_i, c_i = j | \Theta, \Phi^{(t)})} = \frac{\widetilde{\mathbf{F}}_{ik}^{(t)}}{\sum_{j=1}^{K} \widetilde{\mathbf{F}}_{ij}^{(t)}}. \tag{10}$$

The scale factor[36] serving as an intermediate result for subsequent updates in the M-step is :

$$\mathbf{u}_{ik}^{(t+1)} = \frac{2}{1 + D_M(\mathbf{z}_i^{(t)}, \boldsymbol{\mu}_k^{(t)})}. \tag{11}$$

**M-step.** In the M-step of iteration $(t+1)$, given the gradients $\frac{\partial J(\Theta, \Phi)}{\partial \omega_k} = 0$, $\frac{\partial J(\Theta, \Phi)}{\partial \boldsymbol{\mu}_k} = 0$, and $\frac{\partial J(\Theta, \Phi)}{\partial \Sigma_k} = 0$, we derive the analytical solutions for the mixture model parameters $\omega_k$, $\boldsymbol{\mu}_k$, and $\Sigma_k$. Assume the anomalous ratio is $l \in [0, 1]$, the number of the normal samples is $n = \text{int}(l * N)$. The updating process for $\{\omega_k^{(t+1)}, \boldsymbol{\mu}_k^{(t+1)}, \boldsymbol{\Sigma}_k^{(t+1)}\}$ is as follows:

- The mixture weights $\omega_k$ are updated by averaging the posterior probabilities over all data points with the number of samples , reflecting the relative presence of each component in the mixture:

$$\omega_k^{(t+1)} = \sum_{i=1}^{n} \boldsymbol{\tau}_{ik}^{(t+1)} / n. \tag{12}$$

- The prototypes $\boldsymbol{\mu}_k$ are updated to be the weighted average of the data points, where weights are the posterior probabilities:

$$\boldsymbol{\mu}_k^{(t+1)} = \sum_{i=1}^{n} \left( \boldsymbol{\tau}_{ik}^{(t+1)} \mathbf{u}_{ik}^{(t+1)} \mathbf{z}_i \right) / \sum_{i=1}^{n} \left( \boldsymbol{\tau}_{ik}^{(t+1)} \mathbf{u}_{ik}^{(t+1)} \right). \tag{13}$$

- The covariance matrices $\Sigma_k$ are updated by considering the dispersion of the data around the newly computed prototypes:

$$\boldsymbol{\Sigma}_k^{(t+1)} = \frac{\sum_{i=1}^{n} \boldsymbol{\tau}_{ik}^{(t+1)} \mathbf{u}_{ik}^{(t+1)} (\mathbf{z}_i - \boldsymbol{\mu}_k^{(t+1)})(\mathbf{z}_i - \boldsymbol{\mu}_k^{(t+1)})^{\mathsf{T}}}{\sum_{j=1}^{K} \boldsymbol{\tau}_{ij}^{(t+1)}}. \tag{14}$$

#### 3.3.2 Update $\Theta$

We focus on anomaly-aware representation learning and use stochastic gradient descent to optimize the network parameters $\Theta$, by minimizing the following joint loss:

$$\mathcal{L} = -J(\Theta, \Phi) + g(\Theta), \tag{15}$$

where $J(\Theta, \Phi) = \log p(\mathbf{X} | \Theta, \Phi)$. An additional constraint term $g(\Theta)$ is introduced to prevent short-cut solution [15]. In practice, an autoencoder architecture is implemented, utilizing a reconstruction loss $g(\Theta) = \|x - \hat{x}\|^2$ as the constraint.

These updates are iteratively performed until convergence, resulting in optimized model parameters that best fit the given data according to the mixture model framework.

## 4 Experiments

### 4.1 Datasets & Baselines

We evaluated UniCAD on an extensive collection of datasets, comprising 30 tabular datasets that span 16 diverse fields. We specifically focused on naturally occurring anomaly patterns, rather than synthetically generated or injected anomalies, as this aligns more closely with real-world scenarios. The detailed descriptions are provided in Table 4 of Appendix D.1. Following the setup in ADBench [17], we adopt an inductive setting to predict newly emerging data, a highly beneficial approach for practical applications.

To assess the effectiveness of UniCAD, we compared it with 17 advanced unsupervised anomaly detection methods, including: (1) *traditional methods*: SOD [24] and HBOS [16]; (2) *linear methods*: PCA [49] and OCSVM [32]; (3) *density-based methods*: LOF [6] and KNN [38]; (4) *ensemble-based methods*: LODA [39] and IForest [29]; (5) *probability-based methods*: DAGMM [58], ECOD [28], and COPOD [27]; (6) *cluster-based methods*: DBSCAN [13], CBLOF [18], DCOD [45] and KMeans- - [9]; and (7) *neural network-based methods*: DeepSVDD [42] and DIF [51]. These baselines encompass the majority of the latest methods, providing a comprehensive overview of the state-of-the-art. For a detailed description, please refer to Appendix D.2.

### 4.2 Experiment Settings

In the unsupervised setting, we employ the default hyperparameters from the original papers for all comparison methods. Similarly, the UniCAD also utilizes a fixed set of parameters to ensure a fair comparison. For all datasets, we employ a two-layer MLP with a hidden dimension of $d = 128$ and ReLU activation function as both encoder and decoder. We utilize the Adam optimizer [21] with a learning rate of $1e^{-4}$ for 100 epochs. For the EM process, we set the maximum iteration number to 100 and a tolerance of $1e^{-3}$ for stopping training when the objectives converge. The number of components in the mixture model is set as $k = 10$, and the proportion of the outlier is set as $l = 1\%$. We evaluate the methods using Area Under the Receiver Operating Characteristic (AUC-ROC) and Area Under the Precision-Recall Curve (AUC-PR) metrics [17], reporting the average ranking (Avg. Rank) across all datasets. All experiments are run 3 times with different seeds, and the mean results are reported.

### 4.3 Performance and Analysis

**Performance Comparison**. Table 1 presents a comparison of UniCAD with 10 unsupervised baseline methods across 30 tabular datasets using the AUC-ROC metric. The experimental results, which encompass 17 baselines, are included in Tables 5 and 6 of Appendix D.3, with additional experiments on other data domains presented in Appendix E. Our proposed UniCAD achieves the top average ranking, exhibiting the best or near-best performance on a larger number of datasets and confirming advanced capabilities. It is noteworthy that there is no one-size-fits-all unsupervised anomaly detection method suitable for every type of dataset, as demonstrated by the observation that other methods have also achieved some of the best results on certain datasets. However, our model showcased a remarkable ability to generalize across most datasets featuring natural anomalies, as evidenced by statistical average ranking. As for clustering-based methods such as KMeans--, DCOD, and CBLOF, they mostly rank in the top tier among all baseline methods, supporting the advantage of combining deep clustering with anomaly detection. However, our method significantly outperformed these methods by mitigating their limitations and further providing a unified framework for joint representation learning, clustering, and anomaly detection.

**Effectiveness of Vector Sum in Anomaly Scoring**. As demonstrated in Table 1, we compare the anomaly score $\mathbf{o}_i$ derived directly from the generation possibility with its vector summation form $\mathbf{o}_i^V$. According to our statistical findings, we observe that vector scores $\mathbf{o}_i^V$ consistently outperform scalar scores $\mathbf{o}_i$. This indicates that the introduction of the vector summation, analogous to the concept of resultant force, makes a substantial difference in anomaly detection scenarios involving multiple clusters. The performance gains of the vector sum scores strongly demonstrate the effectiveness of the UniCAD in capturing the subtle differences in the distinctions among multiple clusters and underscore the utility of this factor in the context of anomaly detection based on clustering.

Table 1: AUCROC of 10 unsupervised algorithms on 30 tabular benchmark datasets. In each dataset, the algorithm with the highest AUCROC is marked in red, the second highest in blue, and the third highest in green.

| Dataset | OC SVM | LOF | IForest | DA GMM | ECOD | DB SCAN | CBLOF | DCOD | KMeans-- | DIF | UniCAD (Scalar) | UniCAD (Vector) |
|---|---|---|---|---|---|---|---|---|---|---|---|---|
| annthyroid | 57.23 | 70.20 | 82.01 | 56.53 | 78.66 | 50.08 | 62.28 | 55.01 | 64.99 | 66.76 | 75.27 | 72.72 |
| backdoor | 85.04 | 85.79 | 72.15 | 55.98 | 86.08 | 76.55 | 81.91 | 79.57 | 89.11 | 92.87 | 87.28 | 89.24 |
| breastw | 80.30 | 40.61 | 98.32 | N/A | 99.17 | 85.20 | 96.86 | 99.02 | 97.05 | 77.45 | 98.15 | 98.56 |
| campaign | 65.70 | 59.04 | 71.71 | 56.03 | 76.10 | 50.60 | 64.34 | 63.16 | 63.51 | 67.53 | 73.52 | 73.64 |
| celeba | 70.70 | 38.95 | 70.41 | 44.74 | 76.48 | 50.36 | 73.99 | 91.41 | 56.76 | 65.29 | 81.38 | 82.00 |
| census | 54.90 | 47.46 | 59.52 | 59.65 | 67.63 | 58.50 | 60.17 | 72.84 | 63.33 | 59.66 | 67.90 | 67.84 |
| glass | 35.36 | 69.20 | 77.13 | 76.09 | 65.83 | 54.55 | 78.30 | 78.07 | 77.30 | 84.57 | 79.52 | 82.17 |
| Hepatitis | 67.75 | 38.06 | 69.75 | 54.80 | 75.22 | 68.12 | 73.05 | 48.38 | 64.64 | 74.24 | 75.53 | 80.62 |
| http | 99.59 | 27.46 | 99.96 | N/A | 98.10 | 49.97 | 99.60 | 99.53 | 99.55 | 99.49 | 99.53 | 99.52 |
| Ionosphere | 75.92 | 90.59 | 84.50 | 73.41 | 73.15 | 81.12 | 90.79 | 57.78 | 91.36 | 89.74 | 92.04 | 90.37 |
| landsat | 36.15 | 53.90 | 47.64 | 43.92 | 36.10 | 50.17 | 63.69 | 33.40 | 55.31 | 54.84 | 49.60 | 57.37 |
| Lymphography | 99.54 | 89.86 | 99.81 | 72.11 | 99.52 | 74.16 | 99.81 | 81.19 | 100.00 | 83.67 | 99.29 | 99.73 |
| mnist | 82.95 | 67.13 | 80.98 | 67.23 | 74.61 | 50.00 | 79.96 | 65.23 | 82.45 | 88.16 | 86.00 | 86.64 |
| musk | 80.58 | 41.18 | 99.99 | 76.85 | 95.40 | 50.00 | 100.00 | 42.19 | 72.16 | 98.22 | 99.92 | 100.00 |
| pendigits | 93.75 | 47.99 | 94.76 | 64.22 | 93.01 | 55.33 | 96.93 | 94.33 | 94.37 | 93.79 | 95.12 | 95.52 |
| Pima | 66.92 | 65.71 | 72.87 | 55.93 | 63.05 | 51.39 | 71.49 | 72.16 | 70.44 | 67.28 | 75.16 | 74.87 |
| satellite | 59.02 | 55.88 | 70.43 | 62.33 | 58.09 | 55.52 | 71.32 | 55.97 | 67.71 | 74.52 | 72.46 | 77.65 |
| satimage-2 | 97.35 | 47.36 | 99.16 | 96.29 | 96.28 | 75.74 | 99.84 | 86.01 | 99.88 | 99.63 | 99.87 | 99.88 |
| shuttle | 97.40 | 57.11 | 99.56 | 97.92 | 99.13 | 50.40 | 93.07 | 97.20 | 69.97 | 97.00 | 99.15 | 98.75 |
| skin | 49.45 | 46.47 | 68.21 | N/A | 49.08 | 50.00 | 68.03 | 64.34 | 65.47 | 66.36 | 72.26 | 69.69 |
| Stamps | 83.86 | 51.26 | 91.21 | 88.89 | 87.87 | 52.08 | 69.89 | 93.41 | 79.78 | 87.95 | 91.37 | 94.18 |
| thyroid | 87.92 | 86.86 | 98.30 | 79.75 | 97.94 | 53.57 | 94.74 | 78.55 | 92.26 | 96.26 | 97.66 | 97.48 |
| vertebral | 37.99 | 49.29 | 36.66 | 53.20 | 40.66 | 49.74 | 41.01 | 38.13 | 38.14 | 47.20 | 33.11 | 47.37 |
| vowels | 61.59 | 93.12 | 73.94 | 60.58 | 62.24 | 57.50 | 92.12 | 51.56 | 93.45 | 81.02 | 88.38 | 92.09 |
| Waveform | 56.29 | 73.32 | 71.47 | 49.35 | 62.36 | 66.41 | 71.27 | 63.47 | 74.35 | 75.33 | 71.81 | 74.29 |
| WBC | 99.03 | 54.17 | 99.01 | N/A | 99.11 | 87.43 | 96.88 | 94.92 | 97.45 | 81.27 | 97.68 | 98.93 |
| Wilt | 31.28 | 50.65 | 41.94 | 37.29 | 36.30 | 49.96 | 34.50 | 44.71 | 34.91 | 39.46 | 48.95 | 52.56 |
| wine | 73.07 | 37.74 | 80.37 | 61.70 | 77.22 | 40.33 | 27.14 | 82.18 | 27.36 | 41.69 | 82.72 | 95.25 |
| WPBC | 45.35 | 41.41 | 46.63 | 47.80 | 46.65 | 52.22 | 45.32 | 49.67 | 45.01 | 44.69 | 48.02 | 49.90 |
| **Avg. Rank** | 7.8 | 8.9 | 5.1 | 8.7 | 6.4 | 9.3 | 5.7 | 7.4 | 6.0 | 5.8 | 3.7 | 2.6 |

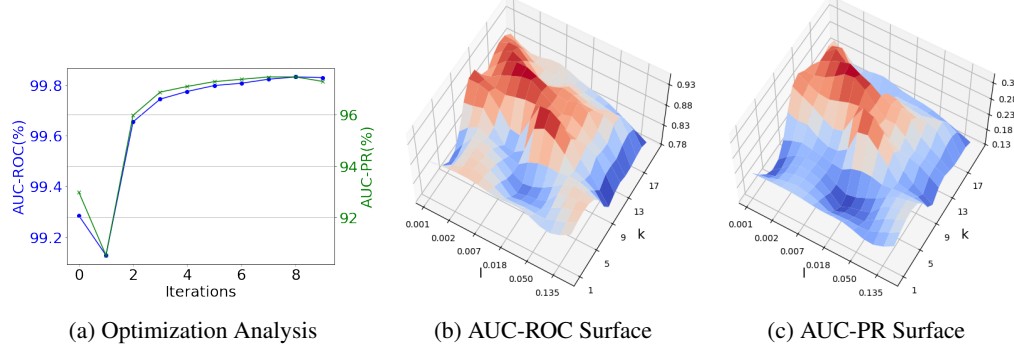

(a) Optimization Analysis     (b) AUC-ROC Surface     (c) AUC-PR Surface

Figure 2: (a) demonstrates the performance variations during the optimization process on the satimage-2 dataset. (b) & (c) Analysis of cluster count $k$, anomaly ratio $l$.

**Analysis of EM Iterative Optimization**. To comprehend the iterative training within our model, we have illustrated the performance variations accompanying the increase in iteration counts in Figure 2a. Specifically, we monitored the iteration number $t$ for the satimage-2 dataset, ranging from 0 to 10, while maintaining other default parameters constant. Both AUC-ROC and AUC-PR performance curves displayed consistent trends, with minor fluctuations only during the initial phase. The performance remained relatively stable throughout the last steps, illustrating the effectiveness and convergence of iterative EM optimization.

**Runtime Comparison.** We present a analysis of the runtime performance of various methods, including our proposed approach, as detailed in Table 2. Our experiments, conducted on the backdoor dataset, reveal that while non-deep learning methods exhibit lower runtime, they often simplify the problem space excessively, failing to capture the complex non-linear relationships present in the data. In contrast, our method, when compared to existing deep learning techniques, demonstrates a significant reduction in computational time. This indicates that our approach not only manages

Table 2: Runtime Comparison. The runtime is reported in seconds (s).

| Phase | IForest | KMeans-- | DAGMM | DCOD | UniCAD |
|-------|---------|----------|--------|----------|--------|
| Fit | 0.256 | 103.697 | 795.004 | 4548.634 | 246.113 |
| Infer | 0.0186 | 0.059 | 4.190 | 16.190 | 0.079 |

Table 3: Ablation study on AUC-ROC scores, calculated across 30 datasets.

| Metric | w/ Gauss. | w/o $J(\Theta, \Phi)$ | w/o $\delta(\mathbf{x}_i)$ | Full Model |
|--------|-----------|------------------------|----------------------------|------------|
| Avg. Rank (w/ baselines & variants) | 6.2 | 6.6 | 5.0 | **4.2** |

to efficiently model complex patterns but also achieves an optimal balance between computational efficiency and modeling capability.

## 4.4 Ablation Studies

In this section, we examine the contributions of different components in UniCAD. Tables 3 reports the results. We make three major observations. **Firstly**, the anomaly detection performance experiences a significant drop when replacing the Student's t distribution with a Gaussian distribution for the Mixture Model, highlighting the robustness of the Student's t distribution in unsupervised anomaly detection. **Secondly**, omitting the likelihood maximization loss (w/o $J(\Theta, \Phi)$) also results in a considerable decrease in overall performance. This observation underscores the importance of deriving both the optimization objectives and anomaly scores from the likelihood generation probability through a theoretical framework, which allows for unified joint optimization of anomaly detection and clustering in the representation space. **Furthermore**, the indicator function $\delta(\mathbf{x}_i)$ also contributes to a performance increase. These results further confirm the effectiveness of our UniCAD in mitigating the negative influence of anomalies in the clustering process, as the existence of outliers may significantly degrade the performance of clustering. In summary, all these ablation studies clearly demonstrate the effectiveness of our theoretical framework in simultaneously considering representation learning, clustering, and anomaly detection.

## 4.5 Sensitivity of Hyperparameters

In this section, we conducted a sensitivity analysis on key hyperparameters of the model applied to the donors dataset, focusing on the number of clusters $k$ and the proportion of the outlier set $l$. The results of this analysis are illustrated in Figure 2. Notably, the optimal range for $l$ tends to be lower than the actual proportion of anomalies in the dataset. Furthermore, a pattern was observed with the number of clusters $k$, where the model performance initially improved with an increase in $k$, followed by a subsequent decline. This suggests the existence of an optimal range for the number of clusters, which should be carefully selected based on the specific application context.

## 5 Conclusion

This paper presents UniCAD, a novel model for Unsupervised Anomaly Detection (UAD) that seamlessly integrates representation learning, clustering, and anomaly detection within a unified theoretical framework. Specifically, UniCAD introduces an anomaly-aware data likelihood based on the mixture model with the Student-t distribution to guide the joint optimization process, effectively mitigating the impact of anomalies on representation learning and clustering. This framework enables a theoretically grounded anomaly score inspired by universal gravitation, which considers complex relationships between samples and multiple clusters. Extensive experiments on 30 datasets across various domains demonstrate the effectiveness and generalization capability of UniCAD, surpassing 15 baseline methods and establishing it as a state-of-the-art solution in unsupervised anomaly detection. Despite its potential, the proposed method's applicability to broader fields like time series and multimodal anomaly detection requires further exploration and validation, highlighting a significant area for future work.

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

---

**Algorithm 1** Model training for UniCAD

---

**Input:** data points $\mathbf{X}$, cluster number $K$, outlier ratio $l$, tolerance $\lambda$, iterations $t$
**Output:** network parameters $\Theta$, mixture parameters $\{\omega_k, \boldsymbol{\mu}_k, \Sigma_k\}$
 1: Initialize $\Theta$ and $\{\boldsymbol{\mu}_k, \omega_k, \Sigma_k\}$;
 2: **for** $i = 1$ to $t$ **do**
 3:     **if** $i = 1$ **then**
 4:         $\mathbf{X}_i \leftarrow \mathbf{X}$;
 5:     **else**
 6:         Re-order the point in $\mathbf{X}$ such that $o_1 \geq \cdots \geq o_n$;
 7:         $L_i \leftarrow \{x_1, \ldots, x_{\lfloor N*l \rfloor}\}$;
 8:         $\mathbf{X}_i \leftarrow \mathbf{X} \setminus L_i$;
 9:     **end if**
10:     Update $\Theta$ with Equation (15);
11:     **while** $|J(\Theta, \Phi) - J^{old}(\Theta, \Phi)| > \lambda$ **do**
12:         $J^{old}(\Theta, \Phi) = J(\Theta, \Phi)$;
13:         Calculate $\boldsymbol{\tau}$ with Equation (10);
14:         Update $\{\omega_k, \boldsymbol{\mu}_k, \Sigma_k\}$ with Equation (12), (13) and (14);
15:     **end while**
16:     Calculate $o_i$ with Equation (9);
17: **end for**
18: **return** $\Theta$ and $\{\omega_k, \boldsymbol{\mu}_k, \Sigma_k\}$

---

## A  Iterative Training Algorithm

The pseudocode for training the model is presented in Algorithm 1. Initially, all parameters undergo random initialization. In subsequent iterations, following the initial round, the outlier set $L$ undergoes updates based on the anomaly score $o_i$. This is succeeded by the adjustment of the network parameters $\Theta$ based on $\mathbf{x}_i$, further optimizing the performance of $\Theta$ through the utilization of the estimated parameters $\boldsymbol{\mu}_k, \omega_k, \Sigma_k$. The essence of the algorithm is embedded in its alternating optimization strategy, iteratively refining the accuracy of representation learning and mixed model parameter estimation, thereby augmenting the overall training effectiveness of the model.

## B  Derivation of EM Algorithm

This appendix provides the detailed derivation of the Expectation-Maximization (EM) algorithm for optimizing the parameters of a mixture model based on Student's t-distribution. The focus is on deriving analytical solutions for the maximization of the parameters $\Phi = \{\boldsymbol{\mu}_k, \Sigma_k, \omega_k\}$ of the mixture components. The EM algorithm alternates between two steps:

**In the E-step**, we calculate the posterior probabilities $\boldsymbol{\tau}_{ik}$, representing the probability of data point $i$ belonging to cluster $k$, given the current parameters. The posterior probabilities for a Student's t-distribution mixture model are formulated as:

$$\boldsymbol{\tau}_{ik} = \frac{\omega_k \cdot p(\mathbf{z}_i | \boldsymbol{\mu}_k, \Sigma_k)}{\sum_{j=1}^{K} \omega_j \cdot p(\mathbf{z}_i | \boldsymbol{\mu}_j, \Sigma_j)}, \tag{16}$$

where $\boldsymbol{\tau}(\mathbf{z}_i | \boldsymbol{\mu}_k, \Sigma_k)$ denotes the Student's t-distribution for data point $i$ with respect to cluster $k$, and $K$ is the number of mixture components.

The Student's t-distribution is depicted as a hierarchical conditional probability, resembling a Gaussian distribution with an accuracy scale factor $\mathbf{u}$, where its latent variable follows a gamma distribution. Adopting a degree of freedom $\nu = 1$, the value of $\mathbf{u}_{ik}$ is given by:

$$\mathbf{u}_{ik} = \frac{\nu + 1}{\nu + D_M(z_i, \boldsymbol{\mu}_k)} = \frac{2}{1 + D_M(z_i, \boldsymbol{\mu}_k)} \tag{17}$$

**In the M-step**, we update the parameters $\Phi = \{\omega_k, \boldsymbol{\mu}_k, \text{and } \Sigma_k\}$ using the derivatives obtained in the previous steps. In our model, the likelihood function for a Student's-t Distribution Mixture Model

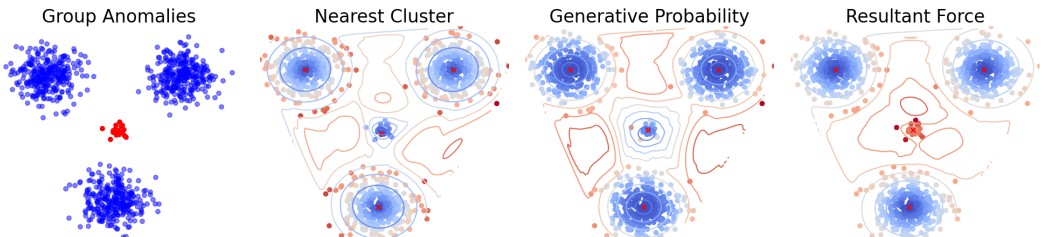

Figure 3: Score comparison with other methods.

(SMM) is represented as:

$$L(\omega, \boldsymbol{\mu}, \Sigma) = \sum_{i=1}^{N} \sum_{k=1}^{K} \omega_k \cdot \frac{\pi^{-1} \cdot |\Sigma_k|^{-\frac{1}{2}}}{1 + (\mathbf{z}_i - \boldsymbol{\mu}_k)^T \Sigma_k^{-1} (\mathbf{z}_i - \boldsymbol{\mu}_k)}, \tag{18}$$

where $\omega_k$ are the mixture weights, $\Sigma_k$ the covariance matrices, $\boldsymbol{\mu}_k$ the means, and $\mathbf{z}_i$ the data points.

The derivative with respect to $\omega_k$ must consider the constraint that the sum of the mixture weights equals 1, i.e., $\sum_k \omega_k = 1$. Hence, we introduce a Lagrange multiplier $\lambda$ to address this constraint and construct the Lagrangian $L'$:

$$L'(\omega, \boldsymbol{\mu}, \Sigma, \lambda) = L(\omega, \boldsymbol{\mu}, \Sigma) + \lambda \left(1 - \sum_{k=1}^{K} \omega_k \right), \tag{19}$$

The derivative with respect to $\omega_k$ is:

$$\frac{\partial L'}{\partial \omega_k} = \frac{\partial L}{\partial \omega_k} - \lambda, \tag{20}$$

Substituting the definition of $L(\omega, \boldsymbol{\mu}, \Sigma)$, we obtain:

$$\frac{\partial L}{\partial \omega_k} = \sum_i \frac{p(\mathbf{z}_i | \boldsymbol{\mu}_k, \Sigma_k)}{\sum_{j=1}^{K} \omega_j \cdot p(\mathbf{z}_i | \boldsymbol{\mu}_j, \Sigma_j)} = \sum_i \frac{\boldsymbol{\tau}_{ik}}{\omega_k}, \tag{21}$$

To solve for $\omega_k$, we first multiply both sides of the equation by $\omega_k$ and apply the constraint condition:

$$\sum_k \omega_k \left( \sum_i \frac{\boldsymbol{\tau}_{ik}}{\omega_k} - \lambda \right) = 0, \tag{22}$$

Upon further organization, we find that the Lagrange multiplier $\lambda$ actually equals the total number of data points $N$ (since $\sum_i \boldsymbol{\tau}_{ik} = N_k$, where $N_k$ is the expected total number of data points belonging to the $k$th component, and the sum of all $N_k$ equals the total number of data points $N$).

Finally, we can solve for $\omega_k$:

$$\omega_k = \frac{\sum_i \boldsymbol{\tau}_{ik}}{N}, \tag{23}$$

This result indicates that the weight $\omega_k$ of each mixture component equals the proportion of the posterior probabilities of the data points it contains relative to all data points.

To update $\boldsymbol{\mu}_k$ and $\Sigma_k$, we consider the conditional expectation of the data log-likelihood function:

$$Q(\boldsymbol{\mu}_k, \Sigma_k) = \sum_{i=1}^{N} \boldsymbol{\tau}_{ik} \left( -\log(\pi) - \frac{1}{2} \log |\sigma_k| + \frac{1}{2} \log u_{ik} \right.$$
$$\left. - \frac{1}{2} \mathbf{u}_{ik} (\mathbf{z}_i - \boldsymbol{\mu}_k)^T \Sigma_k^{-1} (\mathbf{z}_i - \boldsymbol{\mu}_k) \right) \tag{24}$$

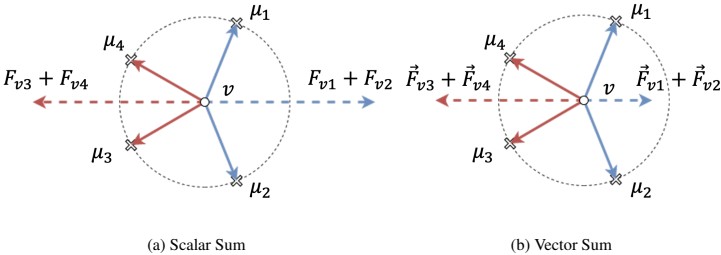

(a) Scalar Sum         (b) Vector Sum

Figure 4: Analysis of gravitational force.

Maximizing $Q(\boldsymbol{\mu}_k, \Sigma_k)$ with respect to $\boldsymbol{\mu}_k$ leads to:

$$\frac{\partial Q}{\partial \boldsymbol{\mu}_k} = \frac{1}{2} \sum_{i=1}^{N} \boldsymbol{\tau}_{ik} \mathbf{u}_{ik} (2\Sigma_k^{-1} \boldsymbol{\mu}_k - 2\Sigma_k^{-1} \mathbf{z}_{ik}) \tag{25}$$

Setting $\frac{\partial Q}{\partial \boldsymbol{\mu}_k} = 0$ results in the updated mean $\boldsymbol{\mu}_k^{(t+1)}$:

$$\boldsymbol{\mu}_k^{(t+1)} = \sum_{i=1}^{n} \left( \boldsymbol{\tau}_{ik}^{(t+1)} \mathbf{u}_{ik}^{(t+1)} \mathbf{z}_i \right) / \sum_{i=1}^{n} \left( \boldsymbol{\tau}_{ik}^{(t+1)} \mathbf{u}_{ik}^{(t+1)} \right). \tag{26}$$

Considering the derivative of $Q(\boldsymbol{\mu}_k, \Sigma_k)$ with respect to $\Sigma_k^{-1}$:

$$\frac{\partial Q}{\partial \Sigma_k^{-1}} = \frac{1}{2} \sum_{i=1}^{N} \boldsymbol{\tau}_{ik} \left( \Sigma_k - \mathbf{u}_{ik}(\mathbf{z}_i - \boldsymbol{\mu}_k) \times (\mathbf{z}_i - \boldsymbol{\mu}_k)^T \right). \tag{27}$$

Setting $\frac{\partial Q}{\partial \boldsymbol{\mu}_k} = 0$ yields the updated covariance matrix $\boldsymbol{\Sigma}_k^{(t+1)}$:

$$\boldsymbol{\Sigma}_k^{(t+1)} = \frac{\sum_{i=1}^{n} \boldsymbol{\tau}_{ik}^{(t+1)} \mathbf{u}_{ik}^{(t+1)} (\mathbf{z}_i - \boldsymbol{\mu}_k^{(t+1)})(\mathbf{z}_i - \boldsymbol{\mu}_k^{(t+1)})^T}{\sum_{j=1}^{K} \boldsymbol{\tau}_{ij}^{(t+1)}}. \tag{28}$$

# C   Anomaly Score with Vector Sum

## C.1   Advantages

Here we discuss the advantages of employing vector sum in anomaly score with a toy example.

The application of the vector sum principle extends beyond physical mechanics and finds relevance in various domains. In relational embedding [5], for example, relationships can be represented as vectors. Aggregating these vectors allows for capturing complexities like transitivity, symmetry, and antisymmetry.

Similarly, in our context, the vector sum can help capture more complex relationships along clusters. Consider Figure 4 as an example, where a sample $v$ is attracted by two groups of cluster prototypes ($\{\boldsymbol{\mu}_1, \boldsymbol{\mu}_2\}$, $\{\boldsymbol{\mu}_3, \boldsymbol{\mu}_4\}$) with the same mass and sample-prototype distances ($\widetilde{m}_1 = \widetilde{m}_2 = \widetilde{m}_3 = \widetilde{m}_4$, $\widetilde{r}_{v1} = \widetilde{r}_{v2} = \widetilde{r}_{v3} = \widetilde{r}_{v4}$). Without considering the direction of the forces, the two groups of prototypes would attract the sample with equal forces. However, we argue that the two groups of prototypes should exert different influences. A sample close to two clusters with a large difference ($\{\boldsymbol{\mu}_1, \boldsymbol{\mu}_2\}$) is more likely to be an anomaly compared to a sample that is close to two clusters with a smaller difference ($\{\boldsymbol{\mu}_3, \boldsymbol{\mu}_4\}$). For example, in a social network, a user who equally likes two extremely different communities, like money-saving tips and luxury items, is more anomalous than a user who equally likes two similar communities, like private jets and luxury items. Applying the vector sum, the total force of $\{\boldsymbol{\mu}_1, \boldsymbol{\mu}_2\}$ is much smaller than that of $\{\boldsymbol{\mu}_3, \boldsymbol{\mu}_4\}$. As the anomaly score is inversely related to the total force, it is more anomalous when equally attracted by $\{\boldsymbol{\mu}_1, \boldsymbol{\mu}_2\}$ with large difference. This indicates that *the vector sum successfully captures subtle differences in the distinctions among multiple clusters, thereby assisting in the identification of more accurate anomalies*.

### C.2 Toy Example

In the appendix, as illustrated in Figure 3, we investigated a toy example. We discussed a specific pattern of anomalies termed *group anomalies*, where a small number of anomalous samples cluster together. It is crucial to note that we do not claim this anomaly pattern is common in real-world data; our goal is merely to point out a specific anomaly pattern that is challenging for traditional cluster-based anomaly detection methods to detect. Specifically, we utilize three Gaussian distributions with high variance (each generating 300 data samples) and one with lower variance (generating 30 data samples). Because the samples from the smaller Gaussian follow a different generative mechanism and represent a minority in the dataset, we consider them anomalies.

We set the cluster number for KMeans-- and GMM at four, indicating that the Gaussian distribution comprising anomalous samples was also recognized as a cluster. KMeans-- employs a cluster-based approach, using the distance to the nearest cluster center as the anomaly score, while GMM uses a probability-based approach, considering the samples' likelihood in the mixture model as the anomaly score. However, both approaches are ineffective in this scenario. Rather than identifying the small cluster as anomalous, they tend to misidentify samples on the peripheries of larger clusters as anomalies.

By contrast, our scoring method views the entire small cluster as more likely anomalous, followed by outlier samples on the margins of the larger clusters. This visualization provides a perspective that distinguishes our method from previous efforts.

## D Experimental Supplementary

### D.1 Benchmark Datasets Details

Due to space constraints in the main text, we utilized 30 public datasets from ADBench [17], covering all different types of data. The details of the 30 datasets are presented in Table 4.

### D.2 Baselines Details

A comprehensive overview of the unsupervised anomaly detection methods is presented below.

#### D.2.1 Traditional Models

- **Subspace Outlier Detection (SOD) [24]:** Identifies outliers in varying subspaces of a high-dimensional feature space, targeting anomalies that emerge in lower-dimensional projections.
- **Histogram-based Outlier Detection (HBOS) [16]:** Assumes feature independence and calculates outlyingness via histograms, offering scalability and efficiency.

#### D.2.2 Linear Models

- **Principal Component Analysis (PCA) [49]:** Utilizes singular value decomposition for dimensionality reduction, with anomalies indicated by reconstruction errors.
- **One-class SVM (OCSVM) [32]:** Defines a decision boundary to separate normal samples from outliers, maximizing the margin from the data origin.

#### D.2.3 Density-based Models

- **Local Outlier Factor (LOF) [6] :** Measures local density deviation, marking samples as outliers if they lie in less dense regions compared to their neighbors.
- **K-Nearest Neighbors (KNN) [38]:** Anomaly scores are assigned based on the distance to the k-th nearest neighbor, embodying a simple yet effective approach.

#### D.2.4 Ensemble-based Models

- **Lightweight On-line Detector of Anomalies (LODA) [39] :** An ensemble method suitable for real-time processing and adaptable to concept drift through random projections and histograms.
- **Isolation Forest (IForest) [29]:** Isolates anomalies by randomly selecting features and split values, leveraging the ease of isolating anomalies to identify them efficiently.

Table 4: Statistics of tabular benchmark datasets.

| Data | # Samples | # Features | # Anomaly | % Anomaly | Category |
|---|---|---|---|---|---|
| annthyroid | 7200 | 6 | 534 | 7.42 | Healthcare |
| backdoor | 95329 | 196 | 2329 | 2.44 | Network |
| breastw | 683 | 9 | 239 | 34.99 | Healthcare |
| campaign | 41188 | 62 | 4640 | 11.27 | Finance |
| celeba | 202599 | 39 | 4547 | 2.24 | Image |
| census | 299285 | 500 | 18568 | 6.20 | Sociology |
| glass | 214 | 7 | 9 | 4.21 | Forensic |
| Hepaitis | 80 | 19 | 13 | 16.25 | Healthcare |
| http | 567498 | 3 | 2211 | 0.39 | Web |
| Ionosphere | 351 | 33 | 126 | 35.90 | Oryctognosy |
| landsat | 6435 | 36 | 1333 | 20.71 | Astronautics |
| Lymphography | 148 | 18 | 6 | 4.05 | Healthcare |
| magic.gamma | 19020 | 10 | 6688 | 35.16 | Physical |
| mnist | 7603 | 100 | 700 | 9.21 | Image |
| musk | 3062 | 166 | 97 | 3.17 | Chemistry |
| pendigits | 6870 | 16 | 156 | 2.27 | Image |
| Pima | 768 | 8 | 268 | 34.90 | Healthcare |
| satellite | 6435 | 36 | 2036 | 31.64 | Astronautics |
| satimage-2 | 5803 | 36 | 71 | 1.22 | Astronautics |
| shuttle | 49097 | 9 | 3511 | 7.15 | Astronautics |
| skin | 245057 | 3 | 50859 | 20.75 | Image |
| Stamps | 340 | 9 | 31 | 9.12 | Document |
| thyroid | 3772 | 6 | 93 | 2.47 | Healthcare |
| vertebral | 240 | 6 | 30 | 12.50 | Biology |
| vowels | 1456 | 12 | 50 | 3.43 | Linguistics |
| Waveform | 3443 | 21 | 100 | 2.90 | Physics |
| WBC | 223 | 9 | 10 | 4. 48 | Healthcare |
| Wilt | 4819 | 5 | 257 | 5.33 | Botany |
| wine | 129 | 13 | 10 | 7.75 | Chemistry |
| WPBC | 198 | 33 | 47 | 23.74 | Healthcare |

### D.2.5 Probability-based Models

- **Deep Autoencoding Gaussian Mixture Model (DAGMM) [58]:** Combines a deep autoencoder with a GMM for anomaly scoring, utilizing both low-dimensional representation and reconstruction error.

- **Empirical-Cumulative-distribution-based Outlier Detection (ECOD) [28]:** Uses ECDFs to estimate feature densities independently, targeting outliers in distribution tails.

- **Copula Based Outlier Detector (COPOD) [27]:** A hyperparameter-free method leveraging empirical copula models for interpretable and efficient outlier detection.

### D.2.6 Cluster-based Models

- **DBSCAN [13]:** A density-based clustering algorithm that identifies clusters based on the density of data points, effectively separating high-density clusters from low-density noise, and is widely used for anomaly detection in spatial data.

- **Clustering Based Local Outlier Factor (CBLOF) [18]:** Calculates anomaly scores based on cluster distances, using global data distribution.

- **KMeans-- [45]:** Extends k-means to include outlier detection in the clustering process, offering an integrated approach to anomaly detection.

- **Deep Clustering-based Fair Outlier Detection (DCFOD) [9]:** Enhances outlier detection with a focus on fairness, combining deep clustering and adversarial training for representation learning.

Table 5: AUCROC of 17 unsupervised algorithms on 30 tabular benchmark datasets. In each dataset, the algorithm with the highest AUCROC is marked in red, the second highest in blue, and the third highest in green.

| Dataset | SOD | HBOS | PCA | OC SVM | LOF | KNN | LODA | IForest | DA GMM | ECOD | COPOD | DB SCAN | CBLOF | DCOD | KMeans-- | Deep SVDD | DIF | UniCAD (Scalar) | UniCAD (Vector) |
|---|---|---|---|---|---|---|---|---|---|---|---|---|---|---|---|---|---|---|---|
| annthyroid | 77.38 | 60.15 | 66.24 | 57.23 | 70.20 | 71.69 | 41.02 | 82.01 | 56.53 | 78.66 | 76.80 | 50.08 | 62.28 | 55.01 | 64.99 | 76.09 | 66.76 | 75.27 | 72.72 |
| backdoor | 68.77 | 71.56 | 80.16 | 85.04 | 85.79 | 80.58 | 66.38 | 72.15 | 55.98 | 86.08 | 80.97 | 76.55 | 81.91 | 79.57 | 89.11 | 78.83 | 92.87 | 87.28 | 89.24 |
| breastw | 93.97 | 98.94 | 95.13 | 80.30 | 40.61 | 97.01 | 98.49 | 98.32 | N/A | 99.17 | 99.68 | 85.20 | 96.86 | 99.02 | 97.05 | 63.36 | 77.45 | 98.15 | 98.56 |
| campaign | 69.16 | 78.55 | 72.78 | 65.70 | 59.04 | 72.27 | 51.67 | 71.71 | 56.03 | 76.10 | 77.69 | 50.60 | 64.34 | 63.16 | 63.51 | 54.42 | 67.53 | 73.52 | 73.64 |
| celeba | 48.44 | 76.18 | 79.38 | 70.70 | 38.95 | 59.63 | 60.17 | 70.41 | 44.74 | 76.48 | 75.68 | 50.36 | 73.99 | 91.41 | 56.76 | 45.17 | 65.29 | 81.38 | 82.00 |
| census | 62.12 | 64.89 | 68.74 | 54.90 | 47.46 | 66.88 | 37.14 | 59.52 | 59.65 | 67.63 | 69.07 | 58.50 | 60.17 | 72.84 | 63.33 | 54.16 | 59.66 | 67.90 | 67.84 |
| glass | 73.36 | 77.23 | 66.29 | 35.36 | 69.20 | 82.29 | 73.13 | 77.13 | 76.09 | 65.83 | 72.43 | 54.55 | 78.30 | 78.07 | 77.30 | 55.71 | 84.57 | 79.52 | 82.17 |
| Hepatitis | 67.83 | 79.85 | 75.95 | 67.75 | 38.06 | 52.76 | 64.87 | 69.75 | 54.80 | 75.22 | 82.05 | 68.12 | 73.05 | 48.38 | 64.64 | 57.45 | 74.24 | 75.53 | 80.62 |
| http | 78.04 | 99.53 | 99.72 | 99.59 | 27.46 | 3.37 | 12.48 | 99.96 | N/A | 98.10 | 99.29 | 49.97 | 99.60 | 99.53 | 99.55 | 60.38 | 99.49 | 99.53 | 99.52 |
| Ionosphere | 86.37 | 62.49 | 79.19 | 75.92 | 90.59 | 88.26 | 78.42 | 84.50 | 73.41 | 73.15 | 79.34 | 81.12 | 90.79 | 57.78 | 91.36 | 53.94 | 89.74 | 92.04 | 90.37 |
| landsat | 59.54 | 55.14 | 35.76 | 36.15 | 53.90 | 57.95 | 38.17 | 47.64 | 43.92 | 36.10 | 41.55 | 50.17 | 63.69 | 33.40 | 55.31 | 62.48 | 54.84 | 49.60 | 57.37 |
| Lymphography | 71.22 | 99.49 | 99.82 | 99.54 | 89.86 | 55.91 | 85.55 | 99.81 | 72.11 | 99.52 | 99.48 | 74.16 | 99.81 | 81.19 | 100.00 | 71.91 | 83.67 | 99.29 | 99.73 |
| mnist | 60.10 | 60.42 | 85.29 | 82.95 | 67.13 | 80.58 | 72.27 | 80.98 | 67.23 | 74.61 | 77.74 | 50.00 | 79.96 | 65.23 | 82.45 | 50.98 | 88.16 | 86.00 | 86.64 |
| musk | 74.09 | 100.00 | 100.00 | 80.58 | 41.18 | 69.89 | 95.11 | 99.99 | 76.85 | 95.40 | 94.20 | 50.00 | 100.00 | 42.19 | 72.16 | 66.02 | 98.22 | 99.92 | 100.00 |
| pendigits | 66.29 | 93.04 | 93.73 | 93.75 | 47.99 | 72.95 | 89.10 | 94.76 | 64.22 | 93.01 | 90.68 | 55.33 | 96.93 | 94.33 | 94.37 | 27.32 | 93.79 | 95.12 | 95.52 |
| Pima | 61.25 | 71.07 | 70.77 | 66.92 | 65.71 | 73.43 | 65.93 | 72.87 | 55.93 | 63.05 | 69.10 | 51.39 | 71.49 | 72.16 | 70.44 | 49.49 | 67.28 | 75.16 | 74.87 |
| satellite | 63.96 | 74.80 | 59.62 | 59.02 | 55.88 | 65.18 | 61.98 | 70.43 | 62.33 | 58.09 | 63.20 | 55.52 | 71.32 | 55.97 | 67.71 | 57.40 | 74.52 | 72.46 | 77.65 |
| satimage-2 | 83.08 | 97.65 | 97.62 | 97.35 | 47.36 | 92.60 | 97.56 | 99.56 | 99.16 | 96.29 | 96.28 | 92.71 | 75.74 | 99.84 | 86.01 | 99.88 | 55.68 | 99.87 | 99.88 |
| shuttle | 69.51 | 98.63 | 98.62 | 97.40 | 57.11 | 69.64 | 60.95 | 99.56 | 97.92 | 99.13 | 99.35 | 50.40 | 93.07 | 97.20 | 69.97 | 51.81 | 97.00 | 99.15 | 98.75 |
| skin | 60.35 | 60.15 | 45.26 | 49.45 | 46.47 | 71.46 | 45.75 | 68.21 | N/A | 49.08 | 47.55 | 50.00 | 68.03 | 64.34 | 65.47 | 45.69 | 66.36 | 72.26 | 69.69 |
| Stamps | 73.26 | 90.73 | 91.47 | 83.86 | 51.26 | 68.61 | 87.18 | 91.21 | 88.89 | 87.87 | 93.40 | 52.08 | 69.89 | 93.41 | 79.78 | 59.48 | 87.95 | 91.37 | 94.18 |
| thyroid | 92.81 | 95.62 | 96.34 | 87.92 | 86.86 | 95.93 | 74.30 | 98.30 | 99.75 | 94.30 |  | 53.57 | 94.74 | 78.55 | 92.26 | 52.14 | 96.26 | 97.66 | 97.48 |
| vertebral | 40.32 | 28.56 | 37.06 | 37.99 | 49.29 | 33.79 | 30.57 | 36.66 | 53.20 | 40.66 | 25.64 | 49.74 | 41.01 | 38.13 | 38.14 | 37.81 | 47.20 | 33.11 | 47.37 |
| vowels | 92.65 | 72.21 | 65.29 | 61.59 | 93.12 | 97.26 | 70.36 | 73.94 | 60.58 | 62.24 | 53.15 | 57.50 | 92.12 | 51.56 | 93.45 | 49.87 | 81.02 | 88.38 | 92.09 |
| Waveform | 68.57 | 68.77 | 65.48 | 56.29 | 73.32 | 73.78 | 60.13 | 71.47 | 49.35 | 62.36 | 75.03 | 66.41 | 71.27 | 63.47 | 74.35 | 53.94 | 75.33 | 71.81 | 74.29 |
| WBC | 94.60 | 98.72 | 98.20 | 99.03 | 54.17 | 90.56 | 96.91 | 99.01 | N/A | 99.11 | 99.11 | 87.43 | 96.88 | 94.92 | 97.45 | 62.46 | 81.27 | 97.68 | 98.93 |
| Wilt | 53.25 | 32.49 | 20.39 | 31.28 | 50.65 | 48.42 | 26.42 | 41.94 | 37.29 | 36.30 | 33.40 | 49.96 | 34.50 | 44.71 | 34.91 | 45.90 | 39.46 | 48.95 | 52.56 |
| wine | 46.11 | 91.36 | 84.37 | 73.07 | 37.74 | 44.98 | 90.12 | 80.37 | 61.70 | 77.22 | 88.65 | 40.33 | 27.14 | 82.18 | 27.36 | 64.26 | 74.69 | 82.72 | 95.25 |
| WPBC | 51.28 | 51.24 | 46.01 | 45.35 | 41.41 | 46.59 | 49.31 | 46.63 | 47.80 | 46.65 | 49.34 | 52.22 | 45.32 | 49.67 | 45.01 | 44.01 | 44.69 | 48.02 | 49.90 |
| **Avg. Rank** | 11.00 | 8.26 | 8.98 | 11.59 | 13.59 | 10.00 | 13.24 | 7.09 | 13.24 | 9.19 | 8.29 | 14.21 | 8.07 | 10.90 | 8.71 | 15.48 | 8.38 | 5.41 | 3.59 |

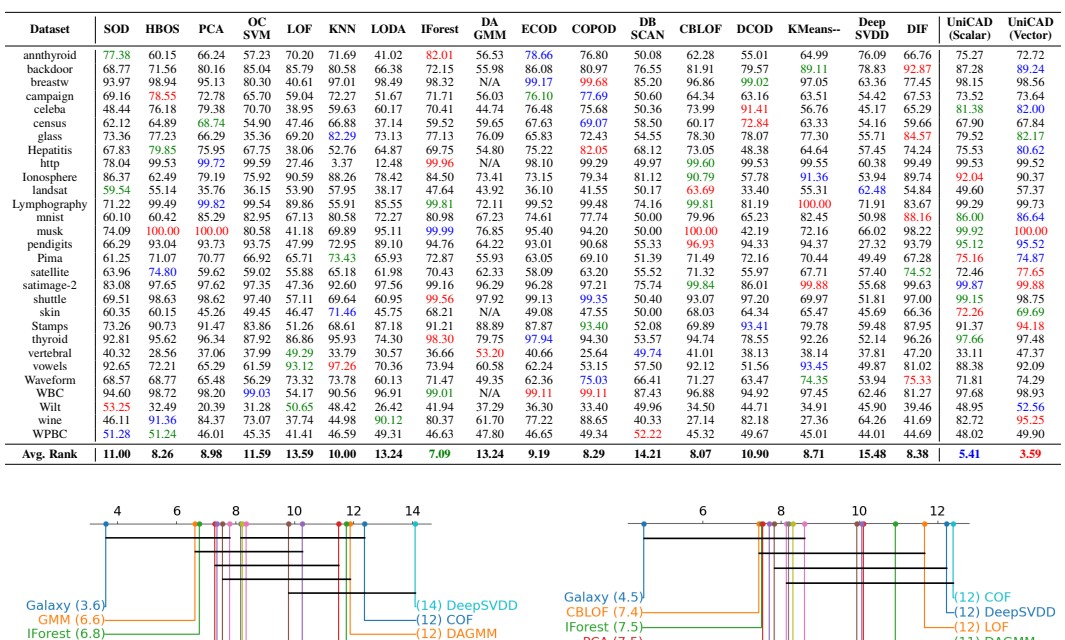

(a) AUC-ROC      (b) AUC-PR

Figure 5: Critical difference diagrams for AUC-ROC and AUC-PR.

#### D.2.7 Neural Network-based Models

- **Deep Support Vector Data Description (DeepSVDD) [42]:** Minimizes the volume of a hypersphere enclosing network data representations, isolating anomalies outside this sphere.

- **Deep Isolation Forest for Anomaly Detection (DIF) [51]:** Utilizes deep learning to enhance traditional isolation forest techniques, offering improved anomaly detection in complex datasets with minimal parameter tuning.

Each method's unique mechanism and application context provide a rich landscape of techniques for unsupervised anomaly detection, illustrating the field's diverse methodologies and the breadth of approaches to tackling anomaly detection challenges.

### D.3 Supplementary Experimental Results

In the appendix, we detail the statistical analysis conducted to compare the performance of various anomaly detectors. We obtained this diagram by conducting a Friedman test (p-value: 4.657e-19), indicating significant differences among different detectors. We utilized average ranks and the Nemenyi test to generate the critical difference diagram, as shown in Figure 5. It is noteworthy that the vector version exhibits significantly superior performance compared to the scalar version across more methods. The detailed outcomes for the AUCROC and AUCPR metrics, spanning 30 datasets and against 17 baseline approaches, are showcased in Table 5 and Table 6.

### D.4 Complexity Analysis

The complexity of each iteration in UniCAD involves three parts: constructing the outlier set, updating the network parameters $\Theta$, and optimizing the mixture model using the EM algorithm.

Table 6: AUCPR of 17 unsupervised algorithms on 30 tabular benchmark datasets. In each dataset, the algorithm with the highest AUCPR is marked in red, the second highest in blue, and the third highest in green.

| Dataset | SOD | HBOS | PCA | OC SVM | LOF | KNN | LODA | IForest | DA GMM | ECOD | COPOD | DB SCAN | CBLOF | DCOD | KMeans-- | Deep SVDD | DIF | UniCAD (Scalar) | UniCAD (Vector) |
|---|---|---|---|---|---|---|---|---|---|---|---|---|---|---|---|---|---|---|---|
| annthyroid | 18.84 | 16.99 | 16.12 | 10.37 | 15.71 | 16.74 | 7.06 | 30.47 | 9.64 | 25.35 | 16.58 | 7.60 | 13.74 | 10.01 | 15.41 | 21.75 | 18.93 | 26.37 | 25.03 |
| backdoor | 37.07 | 4.96 | 31.29 | 8.79 | 26.14 | 44.37 | 13.84 | 4.75 | 5.47 | 10.72 | 7.69 | 21.04 | 7.03 | 6.77 | 15.47 | 55.70 | 41.46 | 37.77 | 36.36 |
| breastw | 84.88 | 97.71 | 95.11 | 82.70 | 28.55 | 92.19 | 97.04 | 96.04 | N/A | 98.54 | 99.40 | 78.42 | 91.94 | 96.83 | 92.25 | 48.60 | 50.65 | 94.47 | 95.90 |
| campaign | 19.14 | 38.01 | 27.90 | 29.25 | 14.59 | 27.18 | 14.11 | 32.26 | 14.54 | 36.65 | 38.58 | 11.43 | 20.88 | 19.61 | 18.86 | 16.75 | 26.52 | 27.66 | 27.12 |
| celeba | 2.36 | 13.82 | 15.89 | 10.73 | 1.73 | 3.14 | 4.04 | 8.96 | 1.95 | 13.96 | 13.69 | 2.32 | 11.22 | 17.48 | 3.19 | 2.73 | 5.44 | 15.12 | 14.66 |
| census | 8.54 | 8.68 | 10.02 | 6.82 | 5.48 | 9.04 | 5.03 | 7.78 | 9.03 | 9.46 | 9.92 | 7.52 | 7.52 | 10.92 | 8.13 | 8.42 | 7.42 | 9.70 | 9.75 |
| glass | 18.73 | 11.82 | 10.05 | 8.02 | 20.11 | 20.26 | 13.37 | 10.99 | 24.58 | 15.35 | 9.78 | 6.88 | 11.57 | 9.66 | 14.66 | 8.46 | 18.86 | 13.29 | 15.33 |
| Hepatitis | 24.73 | 37.73 | 36.65 | 29.44 | 13.67 | 21.95 | 30.90 | 26.25 | 22.93 | 32.80 | 41.50 | 22.31 | 36.54 | 19.53 | 25.14 | 30.04 | 34.93 | 36.08 | 43.37 |
| http | 8.32 | 44.79 | 56.43 | 46.86 | 3.82 | 0.70 | 0.67 | 90.83 | N/A | 16.61 | 35.19 | 0.37 | 47.53 | 44.03 | 45.09 | 13.39 | 41.72 | 43.53 | 43.52 |
| Ionosphere | 85.88 | 41.78 | 73.92 | 74.54 | 88.07 | 90.41 | 73.04 | 80.41 | 64.97 | 64.69 | 69.89 | 63.04 | 89.77 | 47.63 | 91.36 | 43.24 | 87.45 | 89.55 | 87.61 |
| landsat | 26.38 | 22.03 | 16.18 | 16.21 | 24.69 | 24.65 | 18.86 | 19.81 | 24.48 | 16.24 | 17.48 | 20.80 | 31.05 | 15.57 | 22.40 | 36.92 | 24.35 | 20.84 | 23.27 |
| Lymphography | 22.00 | 91.83 | 97.02 | 93.59 | 23.08 | 38.69 | 44.54 | 97.31 | 19.52 | 90.87 | 88.68 | 7.66 | 97.31 | 12.34 | 100.00 | 34.58 | 32.84 | 91.69 | 96.66 |
| mnist | 19.15 | 12.51 | 39.93 | 33.20 | 20.90 | 35.53 | 25.86 | 27.71 | 23.75 | 17.45 | 21.35 | 9.21 | 30.60 | 23.59 | 37.12 | 20.18 | 41.19 | 44.55 | 41.94 |
| musk | 7.59 | 100.00 | 99.89 | 10.61 | 2.82 | 9.65 | 47.60 | 99.61 | 32.76 | 50.13 | 34.79 | 3.16 | 100.00 | 2.87 | 37.55 | 8.78 | 70.70 | 97.65 | 99.96 |
| pendigits | 4.46 | 29.27 | 23.65 | 23.52 | 3.78 | 6.50 | 18.71 | 26.05 | 4.67 | 30.65 | 21.22 | 2.94 | 32.87 | 22.21 | 32.67 | 1.53 | 23.75 | 24.86 | 21.68 |
| Pima | 48.24 | 56.61 | 54.03 | 50.00 | 47.18 | 55.14 | 44.09 | 55.82 | 41.55 | 55.19 | 55.19 | 36.65 | 52.99 | 50.24 | 53.50 | 35.02 | 46.34 | 54.66 | 54.23 |
| satellite | 47.23 | 67.25 | 59.64 | 57.61 | 37.68 | 50.01 | 61.94 | 65.92 | 58.33 | 52.22 | 56.58 | 37.56 | 61.43 | 43.31 | 54.68 | 41.77 | 68.92 | 71.68 | 75.13 |
| satimage-2 | 26.11 | 78.04 | 85.69 | 82.71 | 4.30 | 39.14 | 80.52 | 93.45 | 22.07 | 64.69 | 76.55 | 12.08 | 97.09 | 8.12 | 97.13 | 2.58 | 72.90 | 97.33 | 97.31 |
| shuttle | 20.27 | 96.40 | 92.35 | 85.29 | 13.76 | 20.38 | 48.75 | 97.62 | 93.20 | 90.45 | 96.56 | 7.68 | 79.89 | 81.82 | 32.66 | 12.41 | 67.23 | 92.05 | 92.36 |
| skin | 24.61 | 23.70 | 17.40 | 19.03 | 18.25 | 28.72 | 18.44 | 26.08 | N/A | 18.37 | 17.99 | 20.89 | 28.34 | 26.29 | 25.58 | 19.06 | 25.36 | 28.87 | 28.72 |
| Stamps | 20.28 | 35.24 | 41.09 | 31.39 | 21.29 | 23.53 | 34.60 | 39.49 | 43.73 | 33.21 | 43.10 | 11.03 | 24.46 | 47.36 | 35.63 | 12.07 | 34.68 | 42.39 | 50.94 |
| thyroid | 23.56 | 50.98 | 44.34 | 21.23 | 20.81 | 34.98 | 14.68 | 63.11 | 16.06 | 51.06 | 19.64 | 9.44 | 29.88 | 10.56 | 31.69 | 2.70 | 50.36 | 60.99 | 60.06 |
| vertebral | 11.79 | 9.23 | 10.49 | 10.94 | 14.24 | 10.57 | 9.68 | 10.46 | 15.24 | 11.84 | 8.89 | 13.11 | 11.43 | 11.58 | 10.54 | 10.62 | 14.31 | 9.78 | 12.96 |
| vowels | 38.88 | 13.41 | 8.92 | 8.24 | 34.42 | 63.41 | 13.82 | 15.12 | 12.22 | 10.56 | 4.14 | 13.27 | 35.14 | 3.58 | 49.10 | 4.58 | 14.97 | 26.52 | 32.42 |
| Waveform | 9.66 | 5.86 | 5.79 | 4.37 | 11.33 | 13.04 | 4.71 | 6.24 | 3.11 | 4.76 | 6.90 | 5.33 | 17.93 | 4.26 | 19.74 | 4.41 | 11.28 | 6.49 | 7.83 |
| WBC | 54.00 | 73.56 | 82.29 | 89.87 | 5.57 | 66.55 | 78.67 | 90.49 | N/A | 86.19 | 86.19 | 30.25 | 67.31 | 33.43 | 71.88 | 8.99 | 13.32 | 68.69 | 83.14 |
| Wilt | 5.53 | 3.84 | 3.13 | 3.62 | 5.05 | 4.73 | 3.36 | 4.23 | 4.00 | 3.93 | 3.69 | 5.33 | 3.74 | 4.62 | 3.76 | 4.65 | 4.05 | 4.80 | 5.19 |
| wine | 7.95 | 43.08 | 30.87 | 21.56 | 7.77 | 8.43 | 48.82 | 25.96 | 17.51 | 23.54 | 45.71 | 8.11 | 5.98 | 24.44 | 6.27 | 18.78 | 8.38 | 34.49 | 49.59 |
| WPBC | 25.62 | 23.04 | 23.01 | 22.93 | 20.29 | 21.49 | 25.39 | 22.42 | 22.49 | 21.24 | 22.81 | 23.86 | 21.08 | 22.86 | 20.58 | 25.00 | 20.73 | 22.71 | 24.90 |
| Avg. Rank | 10.83 | 8.19 | 8.31 | 11.14 | 13.24 | 9.36 | 11.79 | 7.29 | 11.96 | 9.36 | 9.53 | 14.91 | 8.53 | 11.97 | 9.03 | 13.41 | 9.10 | 6.31 | 4.74 |

Constructing the outlier set requires a sorting operation, for which we use Numpy's built-in quantile calculation with a time complexity of $\mathcal{O}(N \log N)$. Considering the number of network parameters along with the computation of the loss function, the computational complexity for optimizing $\Theta$ is approximately $\mathcal{O}(TNDd + TNKd)$. The EM algorithm for the Student's t mixture model includes two main steps: the E-step, where the complexity for computing the probability (or responsibility) of each data point belonging to each component is approximately $\mathcal{O}(NKd)$, and the M-step, where the full computational complexity of updating the parameters (mean, covariance matrix) of each component is $\mathcal{O}(NKd^2)$. In practice, we use diagonal covariance matrices, which reduces the update complexity to roughly $\mathcal{O}(NKd)$. If the EM algorithm requires $T$ round to converge, its time complexity is approximately $\mathcal{O}(TNKd)$. Therefore, the time complexity for $t$-iterations is $\mathcal{O}(tN(\log N + Td(D + K)))$.

# E  Additional Experiments on Graph

## E.1  Baselines

Our proposed method was compared with 16 graph domain baseline methods grouped into three categories as follows:

- **Contrastive Learning-based Methods**: This group includes CoLA [30], SLGAD [55], CONAD [53], and ANEMONE [20]. These methods primarily assume that the contrastive loss between anomalous nodes and their neighborhoods is more significant.

- **Autoencoder-based Methods**: This category consists of MLPAE [43], GCNAE [22], DOMI-NANT [11], GUIDE [54], ComGA [31], AnomalyDAE [14], ALARM [37], DONE/AdONE [4] and AAGNN [57]. These methods focus on the reconstruction errors of anomalous nodes during the process of reconstructing the graph structure or features.

- **Clustering-based Methods**: This category of methods encompasses SCAN [52], CBLOF [18], and DCFOD [45]. These methods generally identify anomalies by detecting if a sample deviates from the clustering.

## E.2  Datasets

We assess the performance of our model using four graph benchmark datasets containing organic anomalies. Table 7 presents the statistical summary for each dataset. These datasets contain naturally occurring real-world anomalies and are valuable for assessing the performance of anomaly detection algorithms in real-world scenarios. The sources and compositions of these datasets are as follows:

Table 7: Statistics of graph benchmark datasets.

| Dataset | # Nodes | # Edges | # Features | # Anomaly | Category |
|---------|---------|---------|------------|-----------|----------|
| Disney | 124 | 670 | 28 | 6 | co-purchase network |
| Weibo | 8,405 | 407,963 | 400 | 868 | social media network |
| Reddit | 10,984 | 168,016 | 64 | 366 | user-subreddit network |
| T-Finance | 39,357 | 42,445,086 | 10 | 1,803 | trading network |

- **Weibo**[19] is a labeled graph comprising user posts extracted from the social media platform Tencent Weibo. The user-user graph establishes connections between users who exhibit similar topic labels. A user is considered anomalous if they have engaged in a minimum of five suspicious events, whereas normal nodes represent users who have not.

- **Reddit**[25] consists of a user-subreddit graph extracted from the popular social media platform Reddit. This publicly accessible dataset encompasses user posts within various subreddits over a month. Each user is assigned a binary label indicating whether they have been banned on the platform. Our assumption is that banned users exhibit anomalous behavior compared to regular Reddit users.

- **Disney**[34] is a co-purchase network of movies that includes attributes such as price, rating, and the number of reviews. The ground truth labels, indicating whether a movie is considered anomalous or not, were assigned by high school students through majority voting.

- **T-Finance**[46] aims to identify anomalous accounts within a trading network. The nodes in this network represent unique anonymous accounts, each characterized by ten features related to registration duration, recorded activity, and interaction frequency. Graph edges denote transaction records between accounts. If a node is associated with activities such as fraud, money laundering, or online gambling, human experts will designate it as an anomaly.

## E.3 Experiment Settings

Table 8: AUC-ROC and AUC-PR of 16 unsupervised algorithms on 4 graph benchmark datasets.

| Group | Method | Weibo | | Reddit | | Disney | | T-Finance | |
|-------|--------|---------|--------|---------|--------|---------|--------|-----------|--------|
| | | AUC-ROC | AUC-PR | AUC-ROC | AUC-PR | AUC-ROC | AUC-PR | AUC-ROC | AUC-PR |
| CL-Based | CoLA | 0.382 | 0.087 | 0.527 | 0.036 | 0.455 | 0.060 | 0.243 | 0.031 |
| | SL-GAD | 0.421 | 0.109 | **0.594** | 0.040 | 0.494 | 0.061 | 0.442 | 0.041 |
| | ANEMONE | 0.320 | 0.082 | 0.536 | 0.036 | 0.454 | 0.068 | 0.226 | 0.030 |
| | CONAD | 0.806 | 0.432 | 0.551 | 0.037 | 0.600 | 0.138 | N/A | N/A |
| AE-Based | MLPAE | 0.880 | 0.629 | 0.501 | 0.035 | 0.563 | 0.064 | 0.299 | 0.030 |
| | GCNAE | 0.847 | 0.567 | 0.526 | 0.033 | 0.517 | 0.059 | 0.295 | 0.030 |
| | GUIDE | 0.897 | 0.692 | 0.566 | 0.040 | 0.521 | 0.060 | N/A | N/A |
| | DOMINANT | 0.927 | 0.797 | 0.561 | 0.037 | 0.590 | 0.077 | N/A | N/A |
| | ComGA | 0.925 | 0.809 | 0.568 | 0.037 | 0.494 | 0.058 | N/A | N/A |
| | AnomalyDAE | 0.892 | 0.694 | 0.560 | 0.037 | 0.520 | 0.070 | N/A | N/A |
| | ALARM | 0.952 | 0.843 | 0.559 | 0.037 | 0.595 | 0.123 | N/A | N/A |
| | DONE | 0.856 | 0.579 | 0.551 | 0.037 | 0.517 | 0.061 | 0.550 | 0.046 |
| | AAGNN | 0.804 | 0.530 | 0.564 | **0.045** | 0.479 | 0.059 | N/A | N/A |
| Cluster-Based | SCAN | 0.701 | 0.186 | 0.496 | 0.033 | 0.548 | 0.053 | N/A | N/A |
| | CBLOF* | 0.972 | 0.875 | 0.503 | 0.035 | 0.574 | **0.146** | 0.524 | 0.046 |
| | DCFOD* | 0.684 | 0.196 | 0.552 | 0.038 | 0.675 | 0.119 | 0.521 | 0.066 |
| | UniCAD * | **0.985** | **0.927** | 0.560 | 0.040 | **0.701** | 0.130 | **0.876** | **0.422** |

In this experiment, we compared graph-based methods on relational data. For methods originally designed around feature vectors, including CBLOF, DCFOD, and our approach, we uniformly employed the same graph representation learning technique as described in BGRL [47]. Specifically, we used a two-layer Graph Convolutional Network (GCN) for encoding, which produced output embeddings with a dimensionality of 128. The training epochs were set to 3000, including a warm-up period of 300 epochs. The hidden size of the predictor was set to 512, and the momentum was fixed at 0.99.

## E.4 Performance Analysis

The performance of UniCAD compared to 16 baseline methods on the four datasets are summarized in Table 8. From the results, we have the following observations: Our model consistently outperforms the baseline methods on most datasets, underlining its effectiveness in anomaly detection even within graph data contexts. This highlights the superiority of UniCAD in detecting anomalies in real-world graph data.

When comparing UniCAD with the four contrastive learning-based methods, it exhibits a distinct advantage, outperforming them by a substantial margin across all metrics. Unlike contrastive learning methods that rely on the local neighborhood for anomaly detection, UniCAD leverages the global clustering distribution. This key difference contributes to its consistently superior performance. Although CONAD incorporates human prior knowledge about anomalies, enabling it to outperform other similar methods on the Weibo and Disney datasets, it still falls short compared to our proposed UniCAD.

Compared to the autoencoder-based methods, UniCAD offers the advantage of lower memory requirements along with better performance. Graph autoencoders typically reconstruct the entire adjacency matrix during full graph training, resulting in memory usage of at least $\mathcal{O}(N^2)$. In contrast, UniCAD, as a clustering-based method, only requires $\mathcal{O}(N \times K)$. Among the autoencoder-based methods, GCNAE, DONE, and AdONE can be extended to the T-Finance dataset as they only reconstruct the sampled subgraphs rather than the entire adjacency matrix. However, UniCAD still showcases superior performance while being more memory-efficient.

UniCAD also demonstrates superior performance compared to various other clustering-based methods, including traditional structural clustering (SCAN) methods that treat the embedding from BGRL as tabular data (CBLOF, DCFOD).

