# OpenReview forum: "Towards a Unified Framework of Clustering-based Anomaly Detection"
_NeurIPS.cc/2024/Conference — Submitted to NeurIPS 2024_

### Official Review · Reviewer_R7pC · 2024-07-08

**Soundness:** 3
**Presentation:** 3
**Contribution:** 3
**Rating:** 6
**Confidence:** 4

**Summary:**

The paper addresses unsupervised anomaly detection by proposing a method named UniCAD. The authors aim to enhance anomaly detection performance by establishing a theoretical connection between representation learning, clustering, and anomaly detection. They introduce a unified framework that jointly optimizes these three components, using a probabilistic mixture model and a Student's-t distribution for robust representation learning and clustering. The framework also includes an anomaly-aware data likelihood objective, which reduces the impact of anomalous data on the learning process. Additionally, the authors propose a gravity-inspired anomaly scoring method that leverages relationships between samples and clusters.

**Strengths:**

1. Modeling the connection between representation learning, clustering, and anomaly detection is highly relevant. This paper effectively demonstrates how these three tasks are interrelated and can be jointly optimized to improve anomaly detection performance.

2. The paper is well-written, presenting its hypothesis and method clearly.

3. The results are impressive and demonstrate the effectiveness of UniCAD.

**Weaknesses:**

1. The ablation study on the hyperparameters $k$ and $l$ is insufficient. The authors only present results from a single dataset, satimage-2, where their method achieves an almost perfect score. It would be more informative to perform ablation studies across all 30 datasets or at least a subset where the model also shows lower performance. This broader analysis would demonstrate how these hyperparameters affect the average ranking of the method, similar to the results reported in the paper's table.

2. The authors introduce a $g(\Theta)$ term to prevent shortcut solutions, mentioning it in Equation 15. However, they do not discuss its importance or impact on performance after its introduction. Key questions remain unanswered, such as how the $g(\Theta)$ term affects the model's performance, what happens if it is removed, and how the autoencoder is implemented. These details are crucial, as the regularization term may significantly influence the results.

**Questions:**

See weaknesses.

**Limitations:**

The authors mention some of the limitations, but they do not address the potential negative impact of the work.

---

> ### Author Rebuttal · Authors · 2024-08-07
>
> > The ablation study on the hyperparameters 𝑘 and 𝑙 is insufficient. The authors only present results from a single dataset, satimage-2, where their method achieves an almost perfect score. It would be more informative to perform ablation studies across all 30 datasets or at least a subset where the model also shows lower performance. This broader analysis would demonstrate how these hyperparameters affect the average ranking of the method, similar to the results reported in the paper's table.
>
> Thank you for your valuable feedback. Regarding the ablation study on the hyperparameters (k) and (l), we have tested various values for these parameters and found that the method exhibits a degree of robustness to their specific settings within a reasonable range. As stated in L243, our method utilizes a fixed set of parameters (k=10, l=1%) to ensure a fair comparison. We found that for most datasets, the method performs well with these settings.
>
> Specifically, we conducted a grid search over the following coarse-grained parameter space and compared them across 30 datasets against 17 baseline methods. Indeed, for specific datasets, further tuning of hyperparameters can enhance performance. The detailed results, which represent the average ranking of the method in terms of AUC-ROC, are as follows:
>
> | l\k | 10 | 20 | 30 | 40 |
> | ---- | ---- | ---- | ---- | ---- |
> | 0.01 | 3.34 | 4.31 | 4.69 | 4.71 |
> | 0.05 | 4.44 | 4.23 | 4.65 | 4.88 |
> | 0.10 | 4.27 | 4.46 | 4.48 | 4.88 |
>
> Additionally, we have explored providing guidelines for selecting these hyperparameters. While methods like the elbow method and silhouette coefficient were considered to find the optimal cluster number, they proved time-consuming and not strongly correlated with anomaly detection performance. Instead, an ensemble learning approach, involving random searches of (k) values and aggregating anomaly scores, improved performance on certain datasets and model robustness. We plan to continue this research in future studies.
>
> > The authors introduce a 𝑔(Θ) term to prevent shortcut solutions, mentioning it in Equation 15. However, they do not discuss its importance or impact on performance after its introduction. Key questions remain unanswered, such as how the 𝑔(Θ) term affects the model's performance, what happens if it is removed, and how the autoencoder is implemented. These details are crucial, as the regularization term may significantly influence the results.
>
> Thank you for your valuable feedback regarding the introduction of the term $ g(\Theta) $ in our model. We appreciate your insights and would like to address your concerns as follows:
>
> 1.  Importance and Impact of g(Θ)
>
>
> The constraint term $ g(\Theta) $ is indeed a fundamental and important part of the model. Using only $ J(\Theta, \Phi) $ can lead to shortcut solutions, causing the loss to quickly become infinite, thereby preventing effective optimization of the deep network parameters. As shown in the ablation experiments in Table 3, we aim to demonstrate that having only $ g(\Theta) $ or only $ J(\Theta, \Phi) $ is insufficient; the combination of $ g(\Theta) + J(\Theta, \Phi) $ significantly enhances the model's performance.
>
> 2.  Implementation of the autoencoder
>
>
> Our autoencoder consists of two main components: the encoder and the decoder. The encoder compresses the input data of dimension to a lower-dimensional representation of size. The decoder then converts this compressed representation back to the original data space of dimension. Both the encoder and decoder are implemented using fully connected neural networks with ReLU activation functions. The autoencoder is trained using a mean squared error loss function, which measures the difference between the original input and the reconstructed output.

---

> > ### Comment · Reviewer_R7pC · 2024-08-11
> >
> > I would like to thank the authors for addressing my concerns. I have read the author's responses and all other reviews, and I am keeping my rating the same.

---

> > > ### Author Response · Authors · 2024-08-11
> > >
> > > Thank you very much for reading our rebuttal!
> > > We are very pleased that your concerns have been addressed. We will update the content mentioned in our rebuttal in the camera-ready version.
> > > Thank you again for your valuable suggestions!

---

### Official Review · Reviewer_sbbw · 2024-07-10

**Soundness:** 2
**Presentation:** 3
**Contribution:** 2
**Rating:** 5
**Confidence:** 4

**Summary:**

This paper proposes UniCAD, a theoretically unified framework for representation learning, clustering, and anomaly detection. This paper first introduces the mixture of Student-t distribution $p(x|\Theta, \Phi)$ with degree of freedom $\nu=1$ based on a representation learner $f_\Theta$ using NN. Then, this paper combines with an anomaly indicator $\delta$ for maximum likelihood estimation. Parameters $(\Theta, \Phi)$ are optimized by EM algorithm and SGD. In addition, when detecting anomalies, an improved score is used with reference to gravity. The UniCAD achieved good performance on experiments with various datasets.

**Strengths:**

- This paper is well written and easy to follow.
- Good experimental results.

**Weaknesses:**

- We have several questions about the proposed method and experiments. Please see Qustions.
- The comparison with DeepSVDD and DIF is excellent, but I think the paper also needs to be compared with other Deep anomaly detection methods. For example, DROCC [1].

[1] Goyal, Sachin, et al. "DROCC: Deep robust one-class classification. "International conference on machine learning. PMLR, 2020.

**Questions:**

- if the degrees of freedom is fixed to 1, the benefit of the t-distribution disappears. Why not learn the degrees of freedom as well?
- I think $\tilde{F}_{ik}$ is a scalar and the norm of $\tiled{r}_{ik}$ is 1, so I don't see the difference between Eq. (9) and Eq. (8).
- As shown in Figure 2, I think the proposed method strongly depends on the hyperparameters $l$ and $k$. Is there any criteria for setting these values?
- The method in this paper seems incremental. (Especially until 3.1.) I think the main novelty is the gravity-based anomaly score described in 3.2. To what extent does this anomaly score improve performance compared to the regular anomaly score? Also, can you give a theoretical explanation?


If the above concerns are remedied and a comparison is made between SOTA's Deep Anomaly Detection methodology, I intend to raise the score.

**Limitations:**

- Hyper-parameter sensitivity seems to be one limitation.

---

> ### Author Rebuttal · Authors · 2024-08-07
>
> > The comparison with DeepSVDD and DIF is excellent, but I think the paper also needs to be compared with other Deep anomaly detection methods. For example, DROCC \[1\].\[1\] Goyal, Sachin, et al. "DROCC: Deep robust one-class classification. "International conference on machine learning. PMLR, 2020.
>
> We are grateful for your suggestion to include comparisons with other deep anomaly detection methods, particularly DROCC.
>
> - Although our initial classification included only two NN-based methods, several other methods in different categories also employ representation learning, such as DAGMM and DCFOD. We will revise this classification to avoid any potential misunderstandings.
>
> - We have included an **additional NN-based comparison method (DROCC)** in our experiments. As shown in our global response, our model significantly outperforms it. However, we appreciate your recommendation and will include and compare this method in the revised version.
>
> - Furthermore, as shown in Appendix E, in the experiments on graph data, we compared our method with **13 deep anomaly detection approaches** based on representation learning.
>
> > if the degrees of freedom is fixed to 1, the benefit of the t-distribution disappears. Why not learn the degrees of freedom as well?
>
> Thank you for your insightful comment regarding the fixed degrees of freedom in our model.
>
> Inspired by existing works\[1~2\], the cross-validating of v on the validation set or learning it is optional, especially in the unsupervised setting. However, in our work, we chose to fix it to 1 to simplify the model. This decision was made to reduce complexity and computational overhead while still maintaining robust performance.
>
> | Metric | learn v | fix v=1 |
> | --------------- | ------- | ------- |
> | AUCROC Avg.rank | 4.4 | 3.34 |
> | AUCPR Avg.rank | 5.05 | 4.47 |
>
>
> > I think \tilde{F}\_{ik} 𝑖𝑠 𝑎 𝑠𝑐𝑎𝑙𝑎𝑟 𝑎𝑛𝑑 𝑡ℎ𝑒 𝑛𝑜𝑟𝑚 𝑜𝑓 \tiled{r}\_{ik} is 1, so I don't see the difference between Eq. (9) and Eq. (8).
>
> We appreciate your attention to detail regarding the mathematical formulation. To clarify, Eq. (8) represents a scalar sum, which is a constant addition, while Eq. (9) involves a vector sum where we take the norm of the vector.
>
> The key difference lies in **whether the direction of the sample and the cluster in the representation space is considered**. Clusters in different directions are considered contradictory when estimating the anomaly degree of a sample and will cancel each other out. In Appendix C.1, we also provide an example for intuitive understanding.
>
> > As shown in Figure 2, I think the proposed method strongly depends on the hyperparameters 𝑙 and 𝑘. Is there any criteria for setting these values?
>
> Thank you for your suggestion.
>
> In our experiments, we have tested various values for these parameters and found that **the method exhibits a degree of insensitivity to their specific settings within a reasonable range**. As stated in L243, our method utilizes a fixed set of parameters to ensure a fair comparison (k=10, l=1%). We found that **for most datasets, the method performs well with these settings.**
>
> We have also explored several criteria for selecting these hyperparameters, including the elbow method, silhouette coefficient, and ensemble of random search. We found that they are time-consuming and **not strongly correlated** with anomaly detection performance.
>
> In the future, we will further explore strategies such as Outlier Detection Thresholding \[3\] in future studies.
>
> > The method in this paper seems incremental. (Especially until 3.1.) I think the main novelty is the gravity-based anomaly score described in 3.2. To what extent does this anomaly score improve performance compared to the regular anomaly score? Also, can you give a theoretical explanation?
>
> We appreciate your observation regarding the novelty of the gravity-based anomaly score.
>
> The gravity-based anomaly score represents a significant advancement over traditional anomaly scores due to its ability to **leverage the complex relationships among samples and clusters**. Unlike conventional scores, which often rely on heuristic designs, our approach is **grounded in a theoretical framework** that connects clustering and anomaly detection through posterior probabilities and likelihood estimations. This theoretical underpinning allows our score to more effectively capture the nuances of data distributions, leading to improved anomaly detection performance.
>
> In our comparative experiments presented in Table 1, we discuss the impact of different scoring methods. The results indicate that our gravity-based anomaly score ranks higher on average compared to traditional scores across 30 datasets, demonstrating its **versatility and effectiveness** in diverse scenarios.
>
> Additionally, we provide an intuitive explanation of the gravity-based anomaly score in Appendix C.1, where we illustrate its advantages through a toy example. This example highlights how our score can better identify group anomalies, which are often challenging for traditional methods to detect.
>
> **References:**
>
> \[1\] Laurens Van Der Maaten. Learning a parametric embedding by preserving local structure. In Artificial intelligence and statistics, pages 384–391. PMLR, 2009.
>
> \[2\] Junyuan Xie, Ross Girshick, and Ali Farhadi. Unsupervised deep embedding for clustering analysis. In International conference on machine learning, pages 478–487. PMLR, 2016.
>
> \[3\] Perini L, Bürkner P C, Klami A. Estimating the contamination factor’s distribution in unsupervised anomaly detection\[C\]//International Conference on Machine Learning. PMLR, 2023: 27668-27679.

---

> ### Comment · Reviewer_sbbw · 2024-08-11
> **Thanks for the rebuttal.**
>
> My concerns have been addressed to some extent, especially with the additional experiments involving DROCC. While the method itself appears to be incremental, its performance on tabular data is excellent. I will raise my score.

---

### Official Review · Reviewer_BFMu · 2024-07-11

**Soundness:** 4
**Presentation:** 4
**Contribution:** 3
**Rating:** 7
**Confidence:** 3

**Summary:**

This paper introduces a novel probabilistic mixture model for unsupervised anomaly detection (UAD) that unifies representation learning, clustering, and anomaly detection into a single theoretical framework. The proposed UniCAD model addresses the lack of a unified approach in existing methods, which often consider these components separately or in pairs. The experimental results show that UniCAD consistently outperformed other methods in terms of AUC-ROC and AUC-PR. The model’s iterative optimization process using EM was also highlighted as effective and convergent.

**Strengths:**

- This paper introduces a novel integration of a probabilistic mixture model that unifies representation learning, clustering, and anomaly detection into a single theoretical framework.
- The proposed approach is well-motivated (Fig. 1) and supported by a robust theoretical foundation that maximizes anomaly-aware data likelihood, ensuring the model effectively leverages the interplay between representation learning, clustering, and anomaly detection.
- The paper is well-written, offering clear and comprehensive explanations of the proposed method, including detailed theoretical derivations and intuitive motivations for the design choices. The methodology section is particularly well-structured, logically outlining the steps and equations involved in the proposed model.
- The comprehensive evaluation design underscores the robustness of the proposed method.

**Weaknesses:**

- The connection between force analysis and anomaly detection, particularly between Equations 7 and 8 in Section 3.2.1, could benefit from further justification. While the analogy is interesting, it may not be immediately intuitive for all readers.
- The iterative optimization process may pose scalability issues for large datasets. An in-depth analysis and discussion of this would further strengthen the quality of this research.
- Although the model maps data to a low-dimensional representation space, the effectiveness of this mapping for very high-dimensional datasets could be explored further.

**Questions:**

I have no specific questions.

**Limitations:**

The authors adequately addressed the limitations.

---

> ### Author Rebuttal · Authors · 2024-08-07
>
> Thank you for the feedback. We will address each of the weaknesses and suggestions you mentioned.
>
> > The connection between force analysis and anomaly detection, particularly between Equations 7 and 8 in Section 3.2.1, could benefit from further justification. While the analogy is interesting, it may not be immediately intuitive for all readers.
>
> Thank you for your suggestion! We share similar concerns and, due to space limitations, we have provided **an intuitive explanation** of the advantages of this analogy in Appendix C, along with **a toy example** to illustrate it more vividly. In this example, we offer a detailed explanation of the differences between scalar and vector concepts.
>
> > The iterative optimization process may pose scalability issues for large datasets. An in-depth analysis and discussion of this would further strengthen the quality of this research.
>
> Thank you for your valuable feedback regarding the iterative optimization process and its potential scalability issues for large datasets. We have analyzed the computational complexity in **Appendix D.4.** The time complexity for t iterations is **O(tN(logN + Td(D + K)))**. According to our complexity and run-time analysis, our method is scalable to large datasets.
>
> In the future, we will explore two techniques that can help reduce the computational burden when processing large datasets: 1. Training on multiple manageable-sized data subsets and combining their scores using ensemble methods. 2. The Mini-Batch EM algorithm, which uses only a small batch of the dataset in each iteration.
>
> > Although the model maps data to a low-dimensional representation space, the effectiveness of this mapping for very high-dimensional datasets could be explored further.
>
> Thank you for your suggestion. We agree that this is an important consideration. In fact, we have addressed this issue from two perspectives:
>
> - On one hand, we can directly map high-dimensional data to a lower dimension using deep neural networks, which helps alleviate the problem of high dimensionality.
>
> - Additionally, as shown in Appendix E, we can also easily extend to higher-dimensional graph data, such as the Weibo dataset with 400 feature dimensions, by simply replacing the corresponding backbone, demonstrating competitive performance.

---

> > ### Comment · Reviewer_BFMu · 2024-08-12
> > **Response from Reviewer BFMu**
> >
> > I appreciate the effort the authors put into preparing the rebuttal. I have no further comments and will be keeping my rating as it is.

---

### Official Review · Reviewer_6DdC · 2024-07-11

**Soundness:** 3
**Presentation:** 3
**Contribution:** 3
**Rating:** 6
**Confidence:** 3

**Summary:**

The authors propose UniCAD to jointly model representation learning,
clustering and anomaly detection.  The main objective is maximizing
the product of anomaly indicator (1 is normal, 0 is anomaly) and the
joint probability of instance x_i and cluster c_k given parameters for
representation learning theta and clustering phi.  The joint
probability is decomposed into the prior of c_k and likelihood of
p(x_i|c_k), which is modeled by a Student's-t distribution on the
distance between representation z_i and mean mu_k with covariance
Sigma_k.  p(x_i) is the marginal over c_k.  Anomaly indicator delta is
zero for p(x_i) in the lowest l percentage.  The anomaly score is
1/p(x_i).

Compared to Newton's law of Universal Gravitation, the anomaly score
function has similar components, except for the unit vector r_ik
(which indicates the directions of forces, beyond the
magnitudes). Hence, they incorporated r_ik into their anomaly score
function.

For updating the clustering parameter phi (mixture weights, means,
covariance), they use EM.  In the E-step, they estimate the posterior
p(c_k|x_i).  In the M-step, they estimate phi.  For updating
representation parameters theta, they use gradient descent to minimize
negative log likelihood of instances, together with a reconstruction
loss via an autoencoder to prevent shortcut solutions.

For empirical comparisons, they use 30 tabular data sets and 17
existing algorithms.  The proposed approach generally outperforms the
others in terms of average rank in AUCROC.  The vector version of
anomaly score function is ranked higher than the scalar version.  On
computation time, UniCAD is in the middle among 5 algorithms.  Ablation
studies indicate the contributions of the different components.

**Strengths:**

The main contribution is combining representation learning,
clustering, and anomaly detection in a unified single probabilistic
formulation, which is interesting.

The empirical results indicate that UniCAD compares favorably against
17 existing techniques on 30 tabular datasets.  Compared to four
existing algorithms, computation is not the most intensive.

The paper is generally well written.

**Weaknesses:**

The clustering part is similar to a typical Gaussian mixture model for clustering via EM, except for t-distribution instead of Gaussian and the scaling factor.

Two neural-network-based approaches were compared.  As UniCAD utilizes
representation learning, comparing with more approaches that utilize
representation learning would be significant.  Approaches without
representation learning have an inherent disadvantage.

Some parts could be clarified--see Questions.

**Questions:**

Eq 11: What is the motivation for the scale factor u_ik, used in Eq 13
and 14?

How is K, number of clusters, determined?

Since the method has representation learning, how does the method
prevent the trivial solution of having most/all instances in one
cluster?

Sec 3.2.1: Consider (a simple case of) only two clusters with the same
covariance and "mass", but different centroids.  If an instance is in the middle
between the two clusters, the two r_ik vectors will be in opposite directions,
resulting in an anomaly score of zero.  In gravitational forces, the
two opposite forces cancel out.  However, in anomaly detection, that
might not be desirable, particularly, when the two clusters are far
away.  Any insights?

What is the "schedule" for updating phi and theta?  Updating one to
convergence before updating the other to convergence?

If one "round" is updating phi to convergence and then theta to
convergence?  Are there multiple rounds?  If so, what is the overall
stopping criterion?

**Limitations:**

Limitations of the proposed approach do not seem to be mentioned.

---

> ### Author Rebuttal · Authors · 2024-08-07
>
> We appreciate the reviewers' valuable suggestions for our work. We hope the following responses will clarify any doubts and enhance the quality of our paper.
>
> > The clustering part is similar to a typical Gaussian mixture model for clustering via EM, except for t-distribution instead of Gaussian and the scaling factor.
>
> We acknowledge that the clustering part seems similar to a typical Gaussian Mixture Model (GMM). However, the primary reason we use the Student's t-distribution instead of the Gaussian distribution is **its robustness to outliers**. As mentioned in Section 3.1.1, the introduction of the Student's t-distribution allows the model to perform better when dealing with high-variance data, especially in the presence of outliers. Our ablation study (Table 3) also demonstrates that using the Student's t Mixture Model (SMM) **significantly improves the average performance** of the method compared to GMM.
>
> > Two neural-network-based approaches were compared. As UniCAD utilizes representation learning, comparing with more approaches that utilize representation learning would be significant. Approaches without representation learning have an inherent disadvantage.
>
> Thank you for your valuable suggestion. We also place great importance on comparing our method with various anomaly detection approaches that utilize representation learning.
>
> - We have included **an additional NN-based comparison method** (DROCC) in our experiments. As shown in our general response, our model significantly outperforms it.
>
> - Furthermore, on graph data, we compared our method with **13 deep anomaly detection approaches** based on representation learning.
>
> > Eq 11: What is the motivation for the scale factor u\_ik, used in Eq 13 and 14?
>
> Thank you for your insightful question.
>
> The primary motivation for introducing the scale factor $ u_{ik} $ is to **downweight the influence of outliers in the data when estimating the parameters of the mixture model**. This approach is also grounded in the optimization derivation presented in the paper \[1\].
>
> In the M-step of the EM algorithm, the estimates for the component means $ \mu_i^{(k+1)} $ and covariance matrices $ \Sigma_i^{(k+1)} $ are computed using weighted averages, where the weights are given by $ u_{ij}^{(k)} $. This allows the model to give less importance to observations that are likely to be outliers, thereby improving the robustness of the parameter estimates.
>
> > How is K, number of clusters, determined?
>
> The selection of cluster numbers, in the absence of prior knowledge, is indeed a topic worthy of in-depth discussion. However, according to our hyperparameter analysis, we found that **k=10 generally adapts well to most evaluated datasets.**
>
> Furthermore, we have considered the searching strategy such as elbow method, silhouette coefficient\[2\] and ensemble of random searches in our works, but we found that they are **time-consuming and not strongly correlated with anomaly detection performance**.
>
> We will explore this active direction in future studies.
>
> > Since the method has representation learning, how does the method prevent the trivial solution of having most/all instances in one cluster?
>
> Thank you for highlight this critical question in deep clustering. Indeed, we have some careful design for avoiding this problem.
>
> The key to our model's ability to avoid this issue lies in its **unified optimization objectives** for representation learning and clustering. Using the maximum likelihood of a mixture model as the objective inherently prevents all samples from being assigned to a single cluster, as this would significantly reduce the overall likelihood. Optimizing the maximum likelihood objective leads to **better results with a mixture model compared to a single cluster**. Consequently, the model will naturally distribute samples across multiple clusters.
>
> > Sec 3.2.1: ... in anomaly detection, that might not be desirable, particularly, when the two clusters are far away. Any insights?
>
> Thank you for raising this insightful question. In fact, the case you mentioned is precisely what we consider to be **the most anomalous situation**. Since our anomaly score is **inversely proportional** to the **resultant force**—when the resultant force is minimal, the anomaly score is maximized. As a result, the instance in the situation you described will be recognized as an anomaly with the highest score.
>
> Furthermore, the toy example in Figure 3 illustrates a similar case, showing that the model can effectively detect anomalies situated at the intersection of multiple clusters.
>
> > What is the "schedule" for updating phi and theta? Updating one to convergence before updating the other to convergence?If one "round" is updating phi to convergence and then theta to convergence? Are there multiple rounds? If so, what is the overall stopping criterion?
>
> We appreciate your insightful comments. In practice, we iteratively optimize $\Phi$ and $\Theta$ using tolerance λ and iterations t. Due to space constraints in the main text, more detailed optimization procedures can be found in Algorithm 1 in Appendix A.
>
> - For the parameters $\Phi$ of the mixture model, we adopt a log-likelihood convergence strategy. The algorithm stops when the increase in log-likelihood between iterations falls below a predefined threshold.
>
> - For the parameters $\Theta$ of the deep model, we use a maximum iterations strategy. The algorithm stops after a predefined number of iterations is reached.
>
> **References:**
>
> \[1\] David Peel and Geoffrey J McLachlan. Robust mixture modelling using the t distribution. Statistics and computing, 10:339–348, 2000.
>
> \[2\] Shi C, Wei B, Wei S, et al. A quantitative discriminant method of elbow point for the optimal number of clusters in clustering algorithm\[J\]. EURASIP journal on wireless communications and networking, 2021, 2021: 1-16.

---

> > ### Comment · Reviewer_6DdC · 2024-08-11
> >
> > Thanks for your response
> >
> > > We have included an additional NN-based comparison method (DROCC) in our experiments. As shown in our general response, our model significantly outperforms it.
> >
> > To include more comparisons with methods with representation learning, I suggest reducing the ones without representation learning.
> >
> > > Furthermore, on graph data, we compared our method with 13 deep anomaly detection approaches based on representation learning.
> >
> > Where is that?
> >
> > > Optimizing the maximum likelihood objective leads to better results with a mixture model compared to a single cluster.
> >
> > What is the main reasoning?
> >
> > >  we consider to be the most anomalous situation. Since our anomaly score is inversely proportional to the resultant force.
> >
> > In my simple example, why is an instance between two centrods the most anomalous situation.   Wouldn't an instance very far away from the two centroids be more anomalous?

---

> > > ### Author Response · Authors · 2024-08-11
> > >
> > > > To include more comparisons with methods with representation learning, I suggest reducing the ones without representation learning.
> > >
> > > Thank you for your suggestions. We will incorporate them into the revised version.
> > >
> > > > Where is that?
> > >
> > > We apologize for not clearly stating the location, these results are placed in **Appendix E**.
> > >
> > > > What is the main reasoning?
> > >
> > > Thanks for your further question. The main reason is that the whole dataset consists of several clusters, thus the assumption of mixture model can better fit the dataset distribution than single model. This is also supported by the research such as GMM\[1\]. According to the Approximation Theorem, a mixture model **can approximate any continuous probability density function by increasing the number of distributions**. If all samples were assigned to a single cluster, there **would still be room for improvement in the maximum likelihood solution**.
> > >
> > > As a result, maximizing the likelihood of mixture model can help improve the clustering performance while avoiding the trivial solution that assign all samples into single cluster.
> > >
> > > > In my simple example, why is an instance between two centroids the most anomalous situation. Wouldn't an instance very far away from the two centroids be more anomalous?
> > >
> > > We apologize for the misunderstanding caused by the use of "most" in the previous response. In our newly proposed anomaly score, **both situations you described—an instance between two centroids and an instance far away from the centroids—would be assigned a high anomaly score**.
> > >
> > > In the example you provided, due to the opposing effects of the two centroids, the resultant force $\vec{{\mathbf{F}}}\_{i}$ is smaller, resulting in a higher anomaly score ${o}\_i^V$, meaning that samples with ambiguous category semantics are more likely to be identified as anomalies. Conversely, the probability-based anomaly score fails to detect this type of anomaly, making it less adaptable to a broader range of data.
> > >
> > > **References:**
> > >
> > > \[1\] Reynolds, Douglas A. "Gaussian mixture models." Encyclopedia of biometrics 741.659-663 (2009).

---

> > > > ### Comment · Reviewer_6DdC · 2024-08-13
> > > >
> > > > > The main reason is that the whole dataset consists of several clusters, thus the assumption of mixture model can better fit the dataset distribution than single model.
> > > >
> > > > I think that is for fixed representation (ie, no representation learning).  Consider the simple case of 2 clusters with centroids mu_1 and mu_2.   With representation learning, I think Eq 2 (and hence Eq 1) can be maximized if all instances are at/near mu_1.  That is, one cluster with all instances having roughly the same representation.
> > > >
> > > > > an instance between two centroids and an instance far away from the centroids—would be assigned a high anomaly score.
> > > >
> > > > When an instance is in between two centroids, the resultant force is close to zero and the anomaly score is going toward infinity.  Numerically, in order for an instance that is far away to get a similar anomaly score, the distance needs to go toward infinity.  Hence, I think the resultant force seems to give an instance in between two centroids a much higher anomaly score than I would desire.

---

> > > > > ### Author Response · Authors · 2024-08-13
> > > > >
> > > > > > I think that is for fixed representation (ie, no representation learning). Consider the simple case of 2 clusters with centroids mu_1 and mu_2. With representation learning, I think Eq 2 (and hence Eq 1) can be maximized if all instances are at/near mu_1. That is, one cluster with all instances having roughly the same representation.
> > > > >
> > > > > Thank you for your response. We also noted this important point, **as indicated in section 3.3.2, L220-L222**. When updating the deep network parameters $\Theta$, solely optimizing J(Θ, Φ) can lead to shortcut solutions, causing the loss to quickly become infinite. Therefore, during representation learning, we introduced the Autoencoder reconstruction loss to prevent all samples from being mapped to the same representation.
> > > > >
> > > > >
> > > > > > When an instance is in between two centroids, the resultant force is close to zero and the anomaly score is going toward infinity. Numerically, in order for an instance that is far away to get a similar anomaly score, the distance needs to go toward infinity. Hence, I think the resultant force seems to give an instance in between two centroids a much higher anomaly score than I would desire.
> > > > >
> > > > > Regarding your point about the anomaly score when an instance is between two centroids, we are open to further discussing this scenario. When an instance is located between two centroids and the resultant force is close to zero, its anomaly score indeed becomes higher compared to other types of anomalies. **This situation is also applicable in real-world scenarios.** For example, in identifying **bots (anomalous users)** in social media platforms, their behavior might resemble multiple user groups, making it difficult to classify them into any single cluster, and thus **placing them at the intersections of multiple clusters**. Our model would mark them as potential anomalies. Conversely, users from **niche groups** with unique behaviors might be far from the cluster centers, and our model would **assign them a lower score than the bots**.
> > > > >
> > > > > Considering that in an unsupervised setting, there is no one-size-fits-all anomaly score that can cater to all types of anomalies, it is essential to choose an appropriate metric based on the specific dataset and scenario.

---

### Author Rebuttal · Authors · 2024-08-07

We sincerely appreciate the positive feedback from most reviewers on our paper, as well as the very useful suggestions from different aspects for further improving the quality of our work.

In the rebuttal, we have carefully read the reviews and provided corresponding answers in each individual replies.

We greatly value and appreciate the valuable questions of the reviewers, and we hope to make full use of the discussion phase to engage in in-depth discussions with the reviewers. Therefore, if there are any further suggestions and questions, we sincerely hope the reviewers can bring them up. We will also do our utmost to discuss and further improve our work.

---

### Decision · Program_Chairs · 2024-09-25

**Decision:**

Reject

**Comment:**

This paper presents an unsupervised anomaly detection framework that consists of two components: (i) a framework that jointly learns latent feature representation, data clustering structure and a generative mixture model; (ii) a gravity-inspired anomaly scoring function based on the mixture model output. Experimental results on 30 data sets show this framework consistently achieves the top three performance when compared with 10 baseline methods.

This paper has a plausible initiative and all reviewers commend the extensive and good experimental results. All reviewers found the paper more or less acceptable. However, after screening through the paper and all reviewer-author discussions, I have a strong sentiment of holding back on an "acceptance" recommendation. My main concern is the limited novelty and depth. Below are detailed comments.

-- A reviewer pointed out the limited technical novelty of component (i). Authors did not directly defend and instead emphasized that component (ii) is novel. I agree with the reviewer. Component (i) is basically a standard mixture model based on student-t distribution and latent feature plus a standard alternate optimization process between mixture model learning (using EM) and latent feature learning (using stochastic gradient descent). Based on these, I judge that (i) carries very limited technical novelty.

-- If component (i) alone can boost detection performance, then it can at least be defended as being "useful". However, this is not clear from the paper since the ablation study results in Table 3 do not include performance without component (ii). Interestingly, authors had a perfect opportunity to address this point when a reviewer questioned the impact of (ii) e.g., they could have provided new detection results based on (i) and without (ii). However, authors did not directly respond and instead pointed to Table 1 which compares two variants of (ii) based on (i) -- as they provided new results for many other questions but chose to dodge this one, I have reason to suspect that (i) is not very useful and the superior performance of the proposed framework mainly comes from (ii). This further weakens the contribution of (i).

-- I  noticed components (i) and (ii) run independently, since (i) is about learning a mixture model and (ii) is about scoring anomalies based on (any) mixture model output. Therefore, conceptually, the paper does not really "integrate" latent feature/mixture model learning with anomaly detection, but merely "combine" them -- if the scoring function in (ii) had not been hand-crafted as in the paper but jointly learned with other components in (i), that would be an "integration" and carry more merit. I think "integration" is an oversold concept in this paper.

-- If my above judgements are correct, then the whole merit of this paper is carried by the gravity-inspired anomaly scoring function in Section 3.2. While it does seem interesting and novel, I think it alone is not sufficient for publishing at a top-notch venue like NeurIPS.

Overall, I judge the work has a plausible initiative but falls short on technical novelty and depth. I encourage authors to deepen the work (e.g., get a real "integration" of all components) and resubmit for another round of review.